

# 1 Measurement report: Hygroscopicity of Size-Selective Aerosol
# 2 Particles at Heavily Polluted Urban Atmosphere of Delhi:
# 3 Impacts of Chloride Aerosol

Anil Kumar Mandariya[1,2], Ajit Ahlawat[3], Mohd. M. V. Haneef[1], Nisar A. Baig[1], Kanan
Patel[4], Joshua S. Apte[5], Lea Hildebrandt Ruiz[4], Alfred Wiedensohler[3]*, and Gazala Habib[1]*
[1]Department of Civil Engineering, Indian Institute of Technology Delhi, New Delhi, India
[2]now at: Univ Paris Est Creteil and University Paris Cité, CNRS, LISA, F – 94010 Créteil, France
[3]Leibniz Institute for Tropospheric Research (TROPOS), Permoserstraße, 15 Leipzig, Germany
[4]Department of Civil, Architectural and Environmental Engineering, The University of Texas at Austin, Austin,
Texas, USA
[5]McKetta Department of Chemical Engineering, The University of Texas at Austin, Austin, Texas, USA
*Correspondence to*: Gazala Habib (gazalahabib@civil.iitd.ac.in) and Alfred Wiedensohler (ali@tropos.de)
**Abstract.** Recent studies reveal that wintertime chloride emission in the Delhi region is crucial in governing
enhancement to theoretically calculated aerosol hygroscopicity and aerosol-bound liquid water to trigger Delhi's
fog episodes. Here, we reported that the high volume fractional contribution of ammonium chloride into aerosol
governs the high aerosol hygroscopicity and associated liquid water content based on the experimental data first
time in Delh. The episodically high chlorides bonded with excess ammonia present in Delhi's atmosphere, which
could lead to haze and fog formation under high relative humidity in the region. Therefore, our study suggests
that controlling the plastic-contained waste, open burning, and e-waste industrial chloride emission could
significantly minimize Delhi's heavily polluted haze/fog events. The high chloride (H-Cl) period was observed
significantly ($p<0.05$) higher hygroscopicity ($0.35 \pm 0.06$) compared to high biomass burning (H-BB) ($0.18 \pm$
0.04), high hydrocarbon-like organic aerosol (H-HOA) ($0.17 \pm 0.05$), and relatively cleaner period ($0.27 \pm 0.07$).
In this study, we present the measurement results of bulk aerosol composition of non-refractory PM1 from ACSM
and size-resolved (Nucleation, Aitken, and Accumulated mode particles) hygroscopic growth factor and
associated hygroscopicity parameter at 90% relative humidity (RH) measured using H-TDMA at Delhi Aerosol
Supersite (DAS) first time. The hygroscopic parameter ($\kappa_{\text{H-TDMA\_90\%}}$) was significantly ($p<0.05$) enhanced with
the size of the particles. The observed $\kappa_{\text{H-TDMA\_90\%}}$ ranged from .00 to 0.11 ($0.03 \pm 0.02$), 0.05 to 0.22 ($0.11 \pm$
0.03), 0.05 to 0.30 ($0.14 \pm 0.04$), 0.05 to 0.41 ($0.18 \pm 0.06$), and 0.05 to 0.56 ($0.22 \pm 0.07$) for 20, 50, 100, 150,
and 200 nm aerosol particles, respectively. The Inorganic-to-organic aerosol ratio in aerosol modulated mainly
the aerosol hygroscopicity. In addition, the accumulation mode particle's hygroscopicity was regulated potentially



by the volume fraction of NH₄Cl and OA in aerosol particles. Interestingly, our results reveal that the daytime
flattening pattern of accumulation aerosol particles in diurnal variation is potentially due to counter the effect of
increment of (NH₄)₂SO₄ and NH₄NO₃ and decrement of NH₄Cl and OA in aerosol particles.
**1. Introduction**
The Intergovernmental Panel on Climate Change (IPCC) reported that aerosol-cloud interaction is still not fully
understood and has significant uncertainties in quantifying global radiative budgets. Aerosol hygroscopicity plays
a pivotal role in overcoming and explaining these uncertainties. Hygroscopicity is crucial to understand how the
aerosol particles act as cloud condensation nuclei (CCN) and forms fog droplets/haze at sub-saturated/nearly
saturation and cloud droplets at atmospheric supersaturation levels (McFiggans et al., 2006; Topping and
McFiggans, 2012). Its understanding is crucial to predict the aerosol size distribution and scattering properties
better in global models under varying atmospheric humidity (RH) conditions (Randall et al., 2007).
Hygroscopicity at higher RH atmospheric conditions leads to an enhanced aerosol cross-sectional area, resulting
in efficient light scattering by the aerosol particles (Tang and Munkelwitz, 1994). It mainly depend on particle
size and chemical composition. Generally, the inorganic salts such as ammonium salts of sulfate, nitrate, and
chloride, are highly hygroscopic (Hu et al., 2011; Petters and Kreidenweis, 2007), organic aerosol (OA) are
comparatively less hygroscopic (Jimenez et al., 2009; Kroll et al., 2011), while dust particles and black/elemental
carbon particles are stated as hydrophobic (Seinfeld and Pandis, 2006). Further, the elevated atmospheric RH
during winter and monsoon favour the formation of more oxidized secondary organic aerosol (SOA) via aqueous-
phase (Ervens et al., 2011) and heterogeneous reactions (McNeill, 2015), leads to enhancement in organic aerosol
hygroscopicity (Jimenez et al., 2009; Mei et al., 2013) which adversely impact on the local visibility (Li et al.,
2016; Liu et al., 2012). However, aerosol loading inversely affects aerosol hygroscopicity (Mandariya et al.,
2020a). Apart from it, aerosol loading is also a critical factor in deciding the lifetime of cloud, which affects the
region's rain quantitatively (Albrecht, 1989; Lohmann and Feichter, 2005).
Over the past decades, aerosol hygroscopicity has been intensively measured using hygroscopic tandem
differential mobility analyzer (H-TDMA) (Massling et al., 2005; Gysel et al., 2007; Mandariya et al., 2020;
Swietlicki et al., 2008; Yeung et al., 2014; Kecorius et al., 2019) and CCN (Bhattu and Tripathi, 2015; Gunthe et
al., 2011; Massoli et al., 2010; Ogawa et al., 2016) counter under sub- and supersaturation levels, respectively.
Petters and Kreidenweis (2007) introduced a hygroscopicity parameter, kappa (κ), to associate aerosol
hygroscopicity with its chemical composition. Furthermore, hygroscopicity associated with OA potentially varies



with OA chemical properties like solubility, the extent of dissociation in aerosol water, and surface activity
(Hallquist et al., 2009; Jimenez et al., 2009), leads to difficulty in the quantification of OA hygroscopicity, result
in introducing more discrepancies in predicted and measured aerosol hygroscopicity. Hence, there is a need to
explore measurement-based aerosol hygroscopicity for Delhi's atmosphere to understand the frequent haze/cloud
formations better.
In past decades, fast economic growth and industrialization in the IGP led to severe air quality during wintertime
(Wester et al., 2019). Delhi is potentially affected by local and regional air pollution problems in wintertime (Arub
et al., 2020; Bhandari et al., 2020; Gani et al., 2019; Prakash et al., 2018). Recent studies (Gani et al., 2019; Rai
et al., 2020) have shown chloride is one of the predominant factors to degrade the air quality in the Delhi region
and significantly favour the haze/fog formation during winter (Gunthe et al., 2021). Trash and biomass burning
for heating and waste degradation are among the main contributors to chloride in Delhi (Rai et al., 2020). A recent
study conducted in Delhi reported that frequent high chloride events promotes high aerosol liquid water content
under elevated humid condition leads to haze and poor visibility in the city (Chen et al., 2022). In addition, Gunthe
et al. (2021) showed higher chloride also facilitates enhancement in aerosol hygroscopicity, however, this study
was based on theoretical hygrocopicity. Therefore, it is essential to investigate the impacts of chloride on aerosol
hygroscopicity and its potential to enhance aerosol-bound liquid water based on field measurements. Moreover,
the hygroscopicity of the aerosol particles in the heavily polluted urban atmosphere, which confines to highly
complex composition, is very limited, like Delhi, situated at Indo Gangetic Plain (IGP), India, where air quality
severely degrades during haze/fog-dominated. To the author's best knowledge, the current study is first in Delhi,
India, exploring a complex atmosphere of IGP using H-TDMA-measured aerosol hygroscopicity. Hence, it is
essential to measure size-resolved aerosol hygroscopicity in Delhi's atmosphere and investigate its role in the
context of high chloride.

## 2. Experimental Methods

### 2.1 Aerosol Measurements

Real-time atmospheric aerosol measurements were conducted simultaneously using Hygroscopic-Tandem
Differential Mobility Analyzer (H-TDMA) and Aerodyne Aerosol Chemical Speciation Monitor (ACSM,
Aerodyne Research, Billerica MA) during winter (1st February 2020 to 16th March 2020) at the Indian Institute of
Technology (IIT) Delhi in Block 5, at the height of nearly 15 m. Details on the sampling site can be found in Arub
et al. (2020). In this study, the HTDMA system was used to investigate the hygroscopic growth of size-resolved



particles at 90 % RH. The HTDMA system has been previously used in many field campaigns (Massling et al.,
2007; Wu et al., 2013b; Zhang et al., 2016). The HTDMA system (TROPOS, Germany) is comprised of two
Differential Mobility Analyzers (DMAs, type Hauke-median, TROPOS, Germany), a Condensation Particle
Counter (CPC, Model 3772, TSI Inc., USA) along with a humidifier system located between the two DMAs. The
role of first DMA is to select the quasi-monodisperse particles at a dry diameter ($D$p, dry) with 30% RH. After
that, the size-selected particles pass through a humidity conditioner, which can be adjusted from 30% to 90% RH
by regulating the aerosol and sheath air flow by mixing dry air with RH<5% and humid air with ~95% RH
(Maßling et al., 2003). The uncertainties associated with RH measurement at 90% RH is 1.0%. The particle
hygroscopic growth distribution at dry size ($D$p, dry) at a certain humidity can be easily determined with CPC.
There are two humidity sensors (Vaisala) in the system for aerosol and sheath respectively. The humidity sensors
positioned in the second DMA were calibrated automatically with 100 nm ammonium sulfate (($NH_4$)$_2SO_4$)
particles every 30 min at 90% RH to analyze the stability at high RH. The measurement error of the HTDMA
mainly depends on the uncertainty in measuring and controlling the RH within the system (Su et al., 2010).
Therefore, all RH sensors were calibrated using the Vaisala salt kit comprising LiCl, NaCl, KCl etc. prior the
measurement campaign. Both the DMAs were size calibrated by applying the Latex particles with the standard
size of 200 nm before the start of the measurement. The number concentration peak occurred at 203 nm, referring
to accuracy of DMAs size selection at 1.5%. HTDMA system was operated at 90% RH to measure the hygroscopic
growth factors (HGFs) for particles with $D$p, dry of five different sizes i.e. 20, 50, 100, 150 and 200 nm. The time
resolution of the full scan covering the five sizes was about 30 min.
ACSM was operated at nearly 0.1 lpm at 1 min time resolution in a temperature-controlled laboratory. ACSM
was set to run to measure mass-to-charge ratio (m/z) m/z 10 to m/z 140. The ACSM measures non-refractory
particulate matter less than 1μm (NR-PM$_1$). The concentrate PM$_1$ aerosol beam was impacted on the vaporizer at
600 °C and flash-vaporized compounds were subsequently ionized through impact ionization at 70 eV electron
and detected with a quadrupole mass spectrometer (Ng et al., 2011). The 200 ms amu$^{-1}$ scan speed and pause
setting at 125 for a sampling time (64 s) were set to acquire aerosol mass spectra in ACSM. Detailed operational
procedures for the ACSM are explained elsewhere in Gani et al. (2019).
**2.2 Data Analysis**
**2.2.1 ACSM**



Details on ACSM calibration and data processing are in Patel et al., 2021. We conducted Positive matrix
factorization (PMF) on the data and found a four-factor solution (hydrocarbon-like OA, "HOA"; biomass burning
OA, "BBOA"; less-oxidized OA, "LO-OOA"; more-oxidized OA "MO-OOA) to best represent the data set.
Further details about PMF analysis are in section S.1 of the SI.
The windrose plot was plotted by openair in R package (http://www.r-project.org, http://www.openair-
project.org). The 48-hour back trajectory of air masses reaching Delhi super site (DSL) at 500 m above the ground
at every hour for the entire study period was estimated by an offline based Hybrid Single-Particle Lagrangian
Integrated Trajectory (HYSPLIT4) model developed by NOAA/Air Resources Laboratory (ARL)) (Draxler and
Hess, 1997). The input meteorological data for back trajectories were taken from the Global Data Assimilation
System (GDAS 0.5 degree) archive maintained by ARL (http://ready.arl. noaa.gov/archives.php). Further,
utilizing these estimated back trajectories as input combined with the measured mass fraction of chemical species
of bulk aerosol, Potential Source Contribution Function (PSCF) analysis was carried out with the help of a tool
called Zefir (V 3.7) written in Igor Pro (WaveMetrics). Detail description regarding Zefir tool can be found
elsewhere (Petit et al., 2017). In addition, box plots reported in the subsequent section were also plotted with the
help of this tool. The aerosol liquid water content (ALWC) as a function of inorganic species mass concentration,
ambient temperature (T), and ambient relative humidity (RH), was calculated by ISORROPIA-II model
(Fountoukis and Nenes, 2007).
**2.2.2 H-TDMA**
The humidity sensor of DMA2 was automatically calibrated with 100 nm ammonium sulfate particles after each
scan cycle. Overall, we recorded 1483 H-TDMA scans cycles. Afterward, the % difference between measured
and theoretical growth factors (say $\Delta q$) was calculated after each scan cycle for 100 nm ammonium sulfate
particles. Those scan cycles came between $\Delta q \leq \pm 5\%$, were only carried out for further data treatment, and the
rest scans cycles were discarded (Kecorius et al., 2019). Thus, we had 1102 H-TDMA scan cycles following this
data quality check. Regarding good scan cycles, we had 1449, 1431, 1438, 1470, and 1420 good H-TDMA scans
for 20, 50, 100, 150, and 200 nm particles, respectively, to further analyze. Afterward that, a piecewise linear
TDMAinv algorithm, namely TDMAinv Toolkit, written in IgorPro and developed by Gysel et al. (2009), was
used to do post-data treatment on the raw HGF. Because the measured distribution function is a skewed and
smoothed integral transform of the actual growth factor probability density functions (GF-PDFs). A detailed
description of the raw data processing in the TDMAinv toolkit to measure real HGFs is described in Gysel et al.



(2009). The TDMAinv toolkit was successfully used in various studies around the globe (Gysel et al., 2007; Liu
et al., 2012; Sjogren et al., 2007; Wang et al., 2018a) and at Kanpur, India (Mandariya et al., 2020a). Besides, the
RH in the DMA2 generally achieved the set value of 90% and remained stable within ±1%, although occasionally,
it faced a more considerable drift. All growth factors measured between 88 and 92% RH were corrected to a target
value of 90% (HGF_90%) (Gysel et al., 2007) using the kappa-model suggested in the TDMAinv toolkit (Gysel
et al., 2009) to minimize this DMA2 RH drifts. After it, 979, 957, 972, 969, and 966 scans were found corrected
at target RH for 20, 50, 100, 150, and 200 nm aerosol particles, respectively, which further averaged for 60 min
time resolution and finally, these numbers reached to 425, 429, 419, 424, and 417, respectively.
Further, size-resolved hygroscopicity factors (kappa, κ, say $\kappa_{H\text{-TDMA\_90\%}}$) were calculated from the respective size-
resolved target RH corrected HGFs using equation (1) kappa-Köhler theory (Mandariya et al., 2020a; Petters and
Kreidenweis, 2007).
$$\kappa_{H-TDMA\_90\%} = (HGF\_90\%^3 - 1)\left[\frac{1}{RH} \, exp \, exp \, \left(\frac{4\sigma M_w}{RT\rho_w D_O HGF\_90\%}\right) - 1\right] \quad\quad\quad (1)$$
Where, $\kappa_{H\text{-TDMA\_90\%}}$ is the hygroscopicity factor at 90% RH, HGF_90% is the size-resolve HGF at 90% RH, RH
is the atmospheric relative humidity in fraction, σ is the surface tension of the aerosol liquid droplet-air interface
at the droplet surface in N/m and can be assumed nearly to pure water, R is the universal gas constant in J K$^{-1}$ mol$^{-1}$
, $M_w$ is the molecular mass of water, T is the ambient temperature in Kelvin (K), $\rho_w$ is the density of water in
kg/m$^3$, and $D_o$ is the dry mobility diameter of the particle in m.
**2.2.4 Derived Secondary Inorganic Salts**
The ACSM mainly measures OA, NO$_3$, SO$_4$, NH$_4$, and Cl. Therefore, we adopted a simplified ion-pairing scheme
reported by Gysel et al. (2007). However, Gysel et al. (2007) did not include NH$_4$Cl in their ion-pairing scheme;
therefore, we elaborated this scheme and made some modifications in this scheme to include ACl in the
calculation. Hence, our modified ion-pairing scheme includes ACl, AN, AS, ABS, and SA are shown below:
**Case-1 $R_{SO_4}(NH_4 \, to \, SO_4) \leq 1$**
$SA = 98.0795 \times max(0, (n_S - n_A))$
$ABS = 115.11 \times n_A$
$AS = 0$

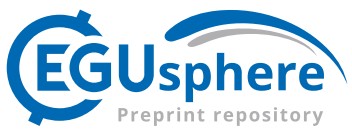

$$AN = \left( \min \left( \left( n_A - \left( \frac{ABS}{115.11} \right) - \left( \frac{(2 \times AS)}{132.1405} \right) \right), n_N \right) \right) \times 80.0434$$
$$ACl = \left( \min \left( n_C, \left( n_A - \left( \frac{ABS}{115.11} \right) - \left( \frac{2 \times AS}{132.1405} \right) - \left( \frac{AN}{80.0434} \right) \right) \right) \right) \times 53.54$$
**Case-2 $1 < R_{SO_4} < 2$**
$SA = 0$
$ABS = 115.11 \times \left( (2 \times n_S) - n_A \right)$
$AS = 132.1405 \times (n_S - n_A)$
$$AN = \left( \min \left( \left( n_A - \left( \frac{ABS}{115.11} \right) - \left( \frac{(2 \times AS)}{132.1405} \right) \right), n_N \right) \right) \times 80.0434$$
$$ACl = \left( \min \left( n_C, \left( n_A - \left( \frac{ABS}{115.11} \right) - \left( \frac{2 \times AS}{132.1405} \right) - \left( \frac{AN}{80.0434} \right) \right) \right) \right) \times 53.54$$
**Case-3 $R_{SO_4} \geq 2$**
$SA = 0$
$ABS = 0$
$AS = 132.1405 \times n_S$
$$AN = \left( \min \left( \left( n_A - \left( \frac{ABS}{115.11} \right) - \left( \frac{(2 \times AS)}{132.1405} \right) \right), n_N \right) \right) \times 80.0434$$
$$ACl = \left( \min \left( n_C, \left( n_A - \left( \frac{ABS}{115.11} \right) - \left( \frac{2 \times AS}{132.1405} \right) - \left( \frac{AN}{80.0434} \right) \right) \right) \right) \times 53.54$$





Here, n denotes the number of moles, whereas A, N, S, and C denotes the $NH_4$, $NO_3$, $SO_4$, and Cl species. We
also predicted these inorganic salts concentrations from the ISORROPIA v2.1 model using $NH_4$, $SO_4$, $NO_3$, and
Cl. We found a strong correlation and nearly unit slope (0.9999) between the calculated and modelled inorganic
salts as presented in figure S1, which strongly justifies the new ion-pairing scheme adopted in this study.
**3. Result and Discussions**
**3.1 Overview of meteorology, trace gases, and aerosol characterization**
Meteorological parameters provided a quick summary of the local weather conditions at the sampling site. Figure
1 shows the hourly resolved temporal variability of meteorological parameters (RH, T, WD, and WS), particle
number size distribution (PNSD), particle volume size distribution (PVSD), principal components in non-
refractory $PM_1$ and OA and their fractional mass contribution. In addition, the temporal variability in the
atmospheric gases (NOx, CO, and $SO_2$) have shown in figure S5. Delhi's winter climate is mainly influenced by
a depression created by the Western Disturbances caused by cold waves in the region. The ambient relative
humidity (RH) and temperature (T) variability between 24.2–96.6% and 9.0–28.5°C with an average (±1 STD) of
$56.0 \pm 18.2\%$ and $18.7 \pm 4.2°C$, respectively, showed the Delhi's atmosphere was varied from wet and cold to dry
and relatively warm from February to March, respectively. The nighttime was somewhat cold and humid
compared to the daytime throughout the sampling. The ambient RH showed a diurnal pattern with a peak during
early morning 06:00-07:00 hr and the valley during noontime 13:00-15:00 hr, while ambient temperature showed
an opposite trend with a rise during noontime could be correlated with noontime higher solar radiation. This
comparatively higher ambient temperature and $O_3$ peak concentration during noontime indicate the daytime
photooxidation process.
The wind speed (WS) and wind direction (WD) varied from 0.0 to 5.6 ($1.0 \pm 1.0$) m/s and 4.0 to 345.7 (197.1 $\pm$
84.4) degrees from the North, respectively, as shown in figure S6. Predominant wind directions were WNW-
WSW and E-ESE. It indicates that the atmosphere remained stagnant during the study period, and measured
aerosol potentially represents Delhi's emissions and local aerosol chemistry.
Furthermore, ambient trace gases NOx and CO showed substantial variability during the sampling period, with a
peak in local burning activities. During intensely biomass burning activities, ambient NOx ambient level reached
a maximum of 421.2 ppb ($58.4 \pm 61.9$). Moreover, the CO showed peak level at similar periods as NOx, and its
concentration varied from 0.0 to 7.66 ppm ($0.58 \pm 0.79$), as shown in figure S5. The diurnal variation of trace
gases is shown in figure 2 (f, g, h, and i). The CO and NOx concentrations showed two peaks in days (06:00-



08:00 and 17:00-20:00), attributed to the morning local biomass/trash burning emissions, night-time traffic rush
hours, and regional biomass burning activities. Besides, $SO_2$ showed a different trend from CO and NOx. $SO_2$
varied dynamically from 0.46 to 9.55 ppb (4.41 ± 1.20) and showed a peak during the morning (09:00-12:00 hr)
and midnight (21:00 to 02:00 hr).
$PM_1$ particle number concentration varied from 408 to 29845 /$cm^3$ (11319 ± 5552). High particle number
concentrations generally observed were associated with local burning events. The particle concentration increased
in the evening (at 18:00 hr) and reached its maximum value at midnight, as shown in figure 2(e). This time
generally indicates the resumption of residential burning activity and traffic emissions. These activities probably
caused the lower geometric mean diameter (GMD) (≈ 47 nm) of the particle number size distribution (PNSD),
which further increased nearly up to 87 nm, as shown in figure 2(t), pointing out the night-time aging of organic
aerosol. The hourly averaged mean diurnal GMD of PVSD varied from nearly 274 to 324 nm, as indicated in
figure 2(y). However, during the sampling period, its variation was observed to vary between 221.3 and 418.4 nm
with the mean value of 309.1 ± 33.1 nm, which is close to this study's higher-end particle size of 200 nm
hygroscopicity measurement. Therefore, ACSM bulk aerosol composition could be the best choice for discussing
the hygroscopicity of 200 nm aerosol particles.
The hourly time-resolved NR-$PM_1$ (say hereafter $PM_1$) concentration varied from 9.0 to 357.9 µg/$m^3$, averaging
81.2 ± 56.6 µg/$m^3$. This observation lies well within the boundary of 12.7-392 µg/$m^3$ (NR-$PM_1$), reported by Gani
et al. (2019) for the current site. Prakash et al. (2018) said that $PM_1$ mass concentration is 83% of $PM_{2.5}$,
representing the dominance of combustion-based particles. Further, we observed that ACSM measured NR-$PM_1$
($PM_1$) was highly correlated ($r^2 = 0.83$, $p<0.05$) with MPSS measured $PM_1$, assuming an effective aerosol density
1.6 g/$cm^3$ (figure S2). It means that a substantial mass of $PM_1$ was composed with non-refractory material and
other refractory material like black carbon, metals and crustal material, which was not measured in the current
study, constituted less than 5% of $PM_1$ (Prakash et al., 2018). OA was the predominant fraction of $PM_1$ with an
average mass concentration of 46.5 ± 39.6 µg/$m^3$, consistent with 112 µg/$m^3$ observed during winter (December-
February) at the present site (Gani et al., 2019). However, lower OA concentration could be explained by the
measuring period of February-March, as aerosol loading starts decreasing in February after reaching its peak in
December-January (Gupta and Mandariya, 2013). Campaign average fractional contribution OA to $PM_1$ was 56%,
ranging from 1 to 84%. This high OA contribution in $PM_1$ is consistent with other studies conducted in IGP
(Chakraborty et al., 2016a; Gani et al., 2019; Mandariya et al., 2019) and worldwide (Jimenez et al., 2009; Zhang
et al., 2007). Peaked OA mass concentrations were noted between 21:00-23:00 hr (figure 2(k)), consistent with



previous studies conducted at the current site (Gani et al., 2019; Rai et al., 2020). Campaign average mass
concentration of $NO_3$ was $10.1 \pm 7.0$ μg/m$^3$ and showed diurnal variation with a peak in the morning and midnight
(figure 2(l)). Besides, $SO_4$ showed slight enhancement at 08:00 hr and remained nearly constant from noon to
17:00 hr (figure 2(m)). However, Cl varied between 0.13 to 77.83 μg/m$^3$, and higher concentrations of Cl were
found episodic throughout the campaign. The Cl- concentration was found consistent with Gani et al., 2019's
previously reported value of 0.1-66.6 μg/m$^3$ at the current site. Figure 1 shows the temporal variation of all
possible OA factors. BBOA mass concentration varied between 0.0 to 134.7 μg/m$^3$. The two peaks during the
night (21:00-23:00 hr) and morning (07:00-08:00 hr) featured the BBOA diurnal variation (figure 2(r)). Besides,
LO-OOA diurnal variation peaked in the morning (10:00 hr) and remained nearly flattered at noontime. It could
indicate its steady formation rate. However, the diurnal variation of MO-OOA showed a slight bump during
noontime, indicating its formation via daytime photooxidation (Mandariya et al., 2019; Sun et al., 2016). Overall,
oxygenated organic aerosol (OOA) was the dominant fraction of OA during the sampling period.
Furthermore, based on the variability in aerosol chemical composition, three different events were observed: first,
high-residential or local biomass burning (H-BB), second, high-hydrocarbon like OA (H-HOA), and third, high-
chloride (H-Cl) period, characterized by high BBOA, HOA, and Cl, respectively. In addition, we also classified
a "Clean Period" where PM$_1$ loading was less than 25 percentile ($\leq 38.7$ μgm$^{-3}$) of the sampling period. H-BB
events showed a dynamic variation in BBOA mass concentration from 16.3 to 134.7 ($50.7 \pm 24.0$) μg/m$^3$.
Although, a higher mass concentration of HOA (9.6 – 109.4 μg/m$^3$) was also observed in these events. It indicates
that HOA also has probably similar sources during this event. Nevertheless, a higher HOA concentration (4.8-
58.9 μg/m$^3$) was noted during H-HOA events, although these concentrations were significantly lower than those
observed during H-BB events. Nevertheless, fractional mass contribution of HOA to OA was largest among all
OA species. Besides, H-Cl events observed a higher concentration of both primary organic aerosol HOA and
BBOA. BBOA contributed nearly 40.0, 21.1, 32.5, and 13.1% to OA during H-BB, H-HOA, H-Cl, and relatively
clean events, respectively, indicating different BBOA sources. Besides, the HOA's average contribution of 41.6%
was observed in the H-HOA event was the highest among all events. In addition, Cl's fractional mass contribution
in PM$_1$ reached up to 44.9% in the H-Cl event compared to 21.2% (H-BB) and 7.3% (H-HOA) events.



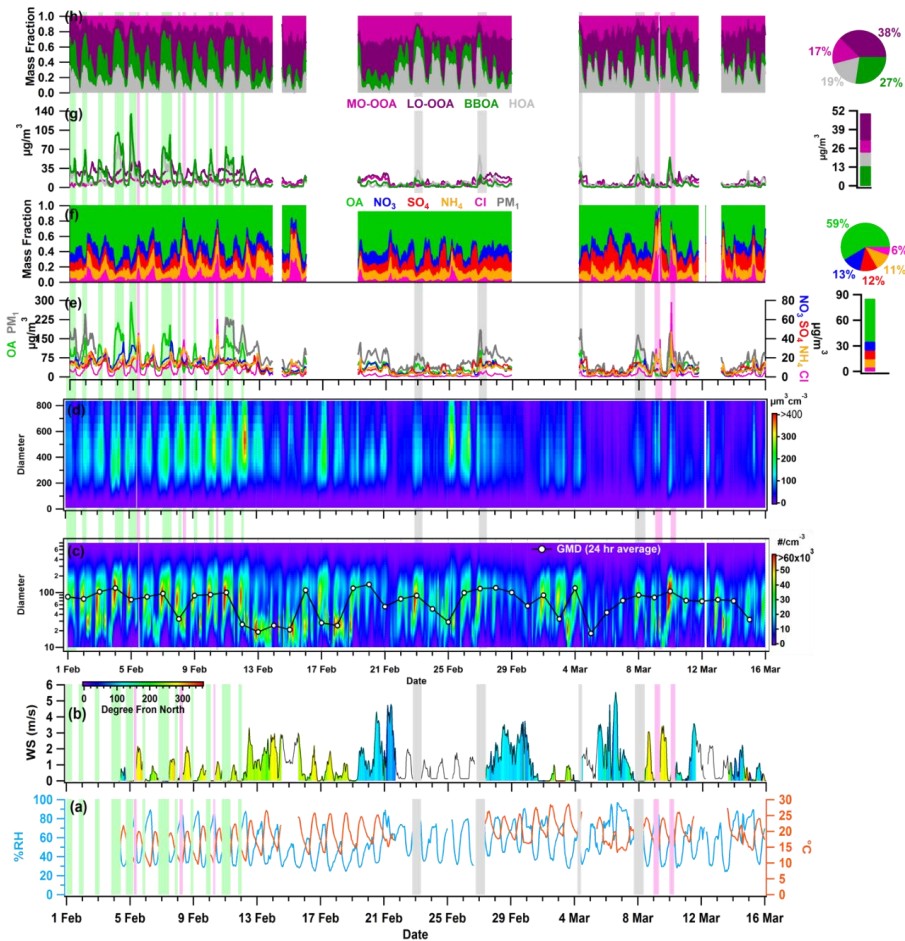

**Figure 1: Temporal variability of ambient (a) relative humidity (RH), temperature (T), (b) wind speed (WS), wind direction (WD), (c) particle number-size distribution (PNSD), 24-average geometric mean diameter (GMD), (d) particle volume-size distribution (PVSD), (e) particulate matter (PM₁), organic aerosol (OA), nitrate (NO₃), sulfate (SO₄), ammonium (NH₄), chloride (Cl), (f) fractional contribution of OA, NO₃, SO₄, NH₄, and Cl in PM₁, (g) more oxidized-oxygenated OA (MO-OOA), less oxidized-oxygenated OA (LO-OOA), biomass burning OA (BBOA), hydrocarbon like-OA (HOA), and (h) fractional contribution of MO-OOA, LO-OOA, BBOA, and HOA in OA. The pie chart sub-plot represents the overall average contribution of species, and the bar sub-plot represents the overall campaign average value of different species. All other species are represented with specific color coding mentioned in legends.**



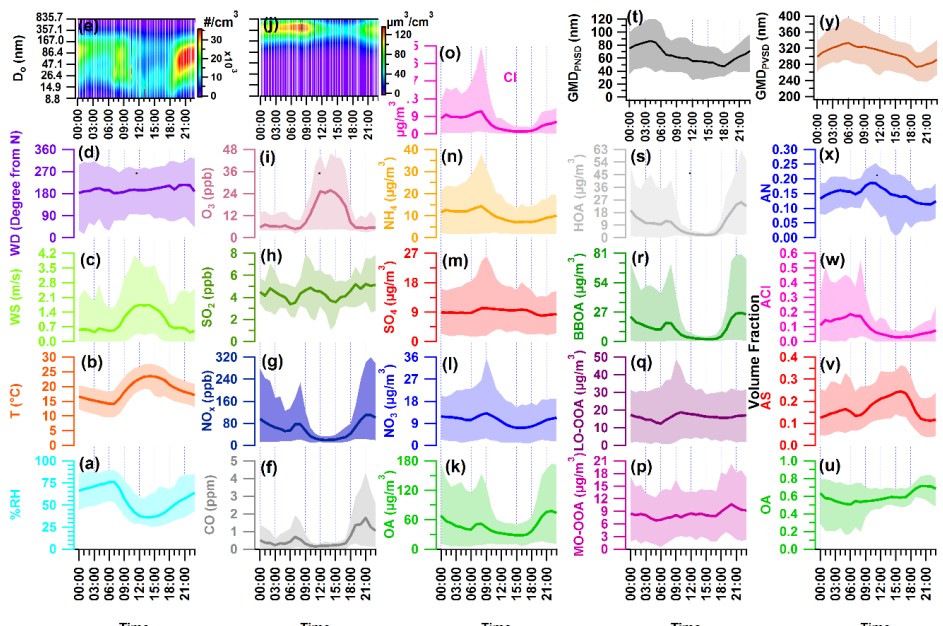

283

**Figure 2: Diurnal variation of ambient meteorological parameters (a) % ambient relative humidity (RH), (b) temperature (T), (c) wind speed (WS), (d) wind direction (WD), and (e) particle number size distribution (PNSD), mass concentration of ambient trace gases (f) carbon mono-oxide (CO), (g) nitrogen oxides (NOx), (h) sulfur dioxide (SO₂), and (i) ozone (O₃), (j) particle volume size distribution (PVSD), mass concentration of aerosol constituents (k) organic aerosol (OA), (l) nitrate (NO₃), (m) sulfate (SO₄), (n) ammonia (NH₄), and (o) chloride (Cl), mass concentration of organic aerosol species (p) more oxidized-oxygenated OA (MO-OOA), (q) less oxidized-oxygenated OA (LO-OOA), (r) biomass burning OA (BBOA), and (s) hydrocarbon like-OA (HOA), (t) geometric mean diameter of particle number size distribution (GMD$_{PNSD}$) and volume fractional contribution of (u) organic aerosol (OA), (v) ammonium sulfate (AS), (w) ammonium chloride (ACl), and (x) ammonium nitrate (AN) in PM₁, and (y) geometric mean diameter of particle volume size distribution (GMD$_{PVSD}$). Upper and lower boundary of shaded area represents the 95th and 5th percentile values of respective species.**

### 3.2 Hygroscopicity of Nucleation, Aitken, and Accumulation Mode Particles

#### 3.2.1 Temporal variability

Figure 3 shows the dynamic variability in the hourly averaged HGF$_{90\%}$ and hygroscopicity parameter ($\kappa_{H\text{-}TDMA\_90\%}$, kappa) of Nucleation, Aitken, and Accumulation mode aerosol particles at 90% ambient relative humidity. The hygroscopic growth factors of 20 (HGF$_{90\%\_20nm}$), 50 (HGF$_{90\%\_50nm}$), 100 (HGF$_{90\%\_100nm}$), 150 (HGF$_{90\%\_150nm}$), and



200 nm (HGF$_{90\%\_200nm}$) size particles varied between 1.00-1.41, 1.05-1.39, 1.11-1.49, 1.12-1.63, and 1.12-1.179
with an average of 1.14 ± 0.09 (average ± standard deviation), 1.16 ± 0.06, 1.27 ± 0.07, 1.35 ± 0.09, and 1.41 ±
0.09, respectively. These mean hygroscopic growth factors were noted as statistically ($p<0.05$) different from each
other. In addition, the hygroscopicity ($\kappa_{20nm\_90\%}$ and $\kappa_{50nm\_90\%}$) of 20 and 50 nm aerosol particles varied between
0.00-0.11 and 0.02-0.25, with an average of 0.03 ± 0.02 and 0.09 ± 0.03, respectively. Nucleation mode particles
were observed, mainly monomodal GF-PDF (figure 3(a)), comprising nearly 74±24% nearly hydrophobic
particles (HGF <1.2). However, this contribution was raised to 100%, which was observed to have a good
association with night-time local burning activities, as shown in the figure 3(a). The nucleation mode particles
($\kappa_{20nm\_90\%}$) showed significantly ($p<0.05$) lower hygroscopicity than Aitken mode particles ($\kappa_{50nm\_90\%}$). Hong et
al. (2015) reported that nucleation mode particles are more sensitive to condensable vapors like fresh VOCs,
$H_2SO_4$ and HCl. However, the present study did not measure these species. The $\kappa$ of Aitken size particles were
comparable with 0.24 ± 0.08 of 52.6 ± 6.9 size particles reported by Gunthe et al. (2011) for Beijing. Beijing is
also one of the most polluted urban locations like Delhi, which could justify the comparison. However, Gunthe et
al. (2011) performed this study using CCN on supersaturation levels. The campaign average hygroscopicity
parameter (kappa, $\kappa_{90\%}$) increased significantly ($p<0.05$) with particle size, which can be attributed to the kelvin
effect (Wang et al., 2018a). In the accumulation size range (100, 150, and 200 nm), $\kappa_{90\%}$ increased to ∼0.56. The
overall sampling average values of $\kappa_{100nm\_90\%}$, $\kappa_{150nm\_90\%}$, and $\kappa_{200nm\_90\%}$ were 0.14 ± 0.04, 0.18 ± 0.06, and 0.22 ±
0.07, respectively. The $\kappa_{200nm\_90\%}$ varied between 0.05 and 0.56. The similar kind of variation in $\kappa$ with particle
size has been demonstrated in Kanpur, situated at the center of IGP, India (Mandariya et al., 2020a) and worldwide
studies (Cerully et al., 2015; Enroth et al., 2018; Fan et al., 2020; Kawana et al., 2016; Kim et al., 2020; Kitamori
et al., 2009; Ogawa et al., 2016; Sjogren et al., 2012; Wang et al., 2018a). Moreover, this was attributed to the
predominant increment in inorganic to OA fraction in particles with increment in size. Furthermore, $\kappa_{H\text{-}TDMA\_90\%}$
was found approximately in the 0.13-0.77, reported by Arub et al. (2020) at Delhi for PM$_1$ without considering
BC. Although, Arub et al. (2020) theoretically predicted particles' hygroscopicity by considering a particle's
chemical composition. They found a decrease in $\kappa$ calculation by 10% when BC was considered in aerosol
chemical composition. Also, $\kappa_{H\text{-}TDMA\_90\%}$ measured in the current study were found in line with the global average
value of 0.27 ± 0.21 for continental aerosols (Petters and Kreidenweis, 2007; Pringle et al., 2010). Further, to
understand the impact of a particle's chemical composition, local meteorology, and air mass trajectories on $\kappa_{H\text{-}}$
$_{TDMA\_90\%}$ for accumulation mode particle discussed in subsequent sections.



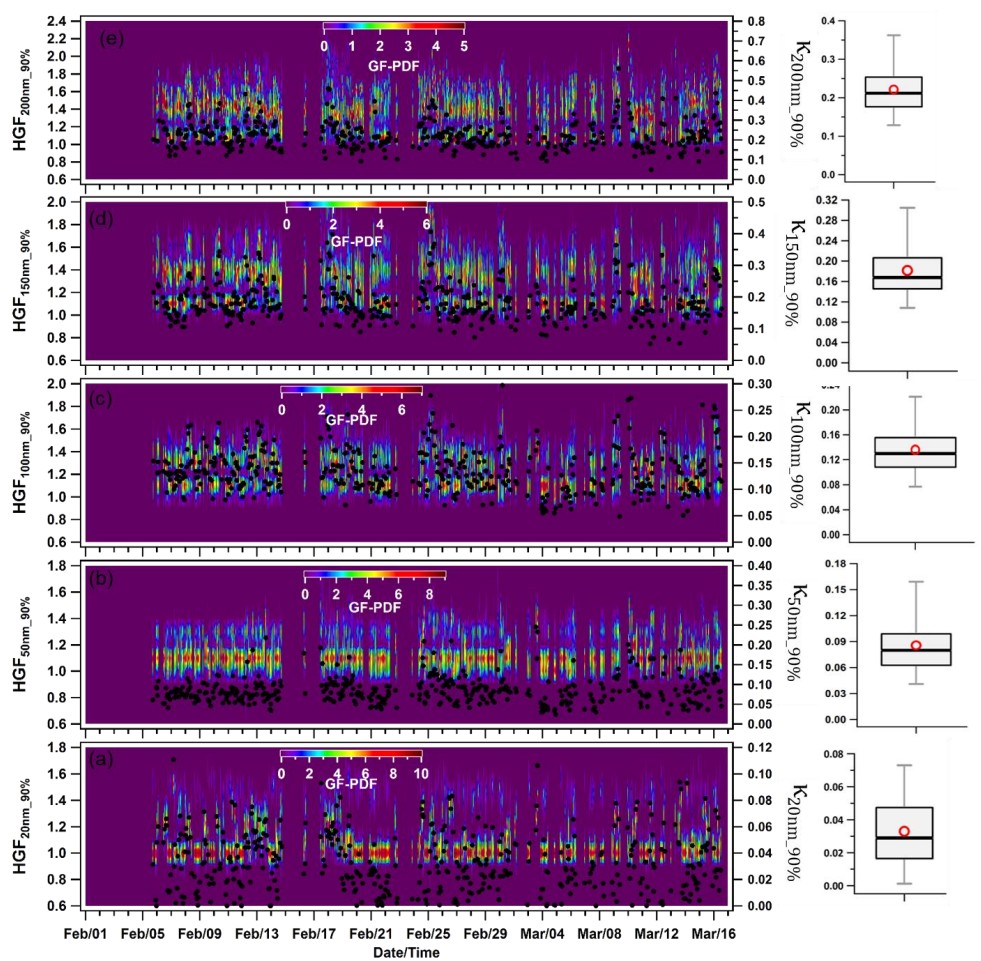

**Figure 3: Temporal variability in hygroscopic parameter kappa ($\kappa$) of nucleation mode particles (a) 20 nm ($\kappa_{20nm\_90\%}$), Aitken mode particles (b) 50 nm ($\kappa_{50nm\_90\%}$), and Accumulation mode particles (c) 100 nm ($\kappa_{100nm\_90\%}$), (d) ($\kappa_{150nm\_90\%}$), and (e) 200 nm ($\kappa_{200nm\_90\%}$). The box plots represent the variability in the hygroscopicity of respective sizes of particles in which low and high whisker traces represent the 5 and 95 percentile, respectively. The red marker indicates the average of the data, whereas the upper and lower sides of the boxes indicate the 75 and 25 percentile of the data, respectively.**

### 3.2.2 Diurnal variability

The diurnal variability in $\kappa_{\text{H-TDMA\_90\%}}$ was found different for nucleation ($\kappa_{20nm\_90\%}$), Aitken ($\kappa_{50nm\_90\%}$), and Accumulation ($\kappa_{100nm\_90\%}$, $\kappa_{150nm\_90\%}$, and $\kappa_{200nm\_90\%}$) mode particles. Figure 4 displayed a diel variation of an average of hourly-resolved $\kappa$ for each size. The bigger size particles exhibited higher values of $\kappa$ than smaller size



particles, which is a similar trend reported at Kanpur, India (Mandariya et al., 2020a) and other worldwide
locations (Fan et al., 2020; Hong et al., 2015). In general, it was observed that all size particles exhibited late-
night hump (02:00-05:00 hr) in $\kappa_{\text{H-TDMA\_90\%}}$. Besides, only $\kappa_{\text{20nm\_90\%}}$ demonstrated a clear diurnal variability with
two peaks, one late night (02:00-04:00 hr) and the other in noontime (14:00-16:00 hr), and two valleys during the
morning (07:00-10:00 hr) and night (19:00-22:00 hr). These valleys reflects the strong impacts of local burning
and traffic activities (Pringle et al., 2010). In addition, nucleation size particles were potentially contributed by
nearly hydrophobic particles (HGF<1.2) from evening to midnight. They showed mono modal GF-PDF around
unit hygroscopic growth factor, possibly indicating local emission generated particles.  The 20 nm particles are
small enough and lie on the boundary of nucleation mode particles. Achtert et al. (2009) reported a similar diurnal
trend of Nucleation and Aitken mode particles, attributed the lower values to the emission of hydrophobic aerosol
particles during the local burning emissions. Daytime hump is attributed to the intense photochemical oxidation
process, which causes the enhancement of more oxidized species on the aerosol particle. Furthermore, their
chemical composition is dominantly controlled by the gaseous condensation of $H_2SO_4$, $HNO_3$, and VOCs (Hong
et al., 2015). The aerosol's chemical composition can address this variability of $\kappa_{\text{H-TDMA\_90\%}}$. However, $\kappa_{\text{50nm\_90\%}}$
also follows a similar diurnal variability as $\kappa_{\text{20nm\_90\%}}$, although it showed less variability. Further, as the dry size
of the aerosol particles increased to accumulated mode, diurnal variation shifted toward nearly steady for the rest
of the day. Hong et al. (2018) also observed no obvious diurnal pattern for 100 and 150 nm particles of organic-
dominated aerosols over the Pearl River Delta region in China.
Furthermore, the diurnal cycles of aerosol physicochemical properties also reflect the dynamic diurnal variation
in the planetary boundary layer (PBL) that leads to the accumulation of particles during night-time. Although this
study did not quantify size-resolved chemical composition, so, this study used bulk-aerosol composition to address
the trend variability only. However, daily average aerosol PNSD varied between 18.0-140.0 nm with a mean of
73.1 ± 33.8 nm. And, the mode of PVSD changed approximately around 300-600 nm. Therefore, it could be an
excellent approximation to discuss $\kappa_{\text{200nm\_90\%}}$ variability with aerosol's bulk chemical properties. The midnight to
early morning hump in hygroscopicity of accumulation mode particles can be attributed to the high rise in the
ratio of inorganic volume fraction to OA volume fraction (Fan et al., 2020), as illustrated in figure 2 (r, s t, u, and
v). Moreover, during mid-night and early morning in the winter, water-soluble organic and inorganic gases are
partitioned and/or coagulated/condensed on the surface of the pre-existing particles. Further, in the presence of
high RH and lower temperature, primary and secondary less oxidized organic aerosol participated in the aging
process, which leads to enhancement their oxidation via aqueous/heterogeneous reaction, according to it increase





the particle's hygroscopicity (Jimenez et al., 2009; Wu et al., 2016). Similar results were observed by Fan et al.
(2020) during winter in urban Beijing, and they attributed it with the enhancement of more hygroscopic particles
due to the aqueous-oxidation and/or condensation process on the pre-existing particles. In general, higher
noontime solar radiation favors more intense photooxidation processes. It supports the partitioning of relatively
more oxidized and less volatile organics on the particulate surface, enhancing the hygroscopicity of accumulation
mode particles (Duplissy et al., 2011; Massoli et al., 2010; Tritscher et al., 2011). However, interestingly, we
observed a noontime flatten pattern of $\kappa_{H\text{-}TDMA\_90\%}$, and it could be attributed to the mix of the positive and negative
impact of an enhancement in the volume fraction of OA and more hygroscopic ammonium sulfate and decrement
in ACl, and AN's volume fraction. Lower volume fractional contribution of highly volatile ACl could be the
potential factor that modulates accumulation mode particle's hygroscopicity. This can be supported by the strong
correlation of $\kappa_{H\text{-}TDMA\_90\%}$ and volume fraction of ACl in that size particles ($\varepsilon_{ACl}$).

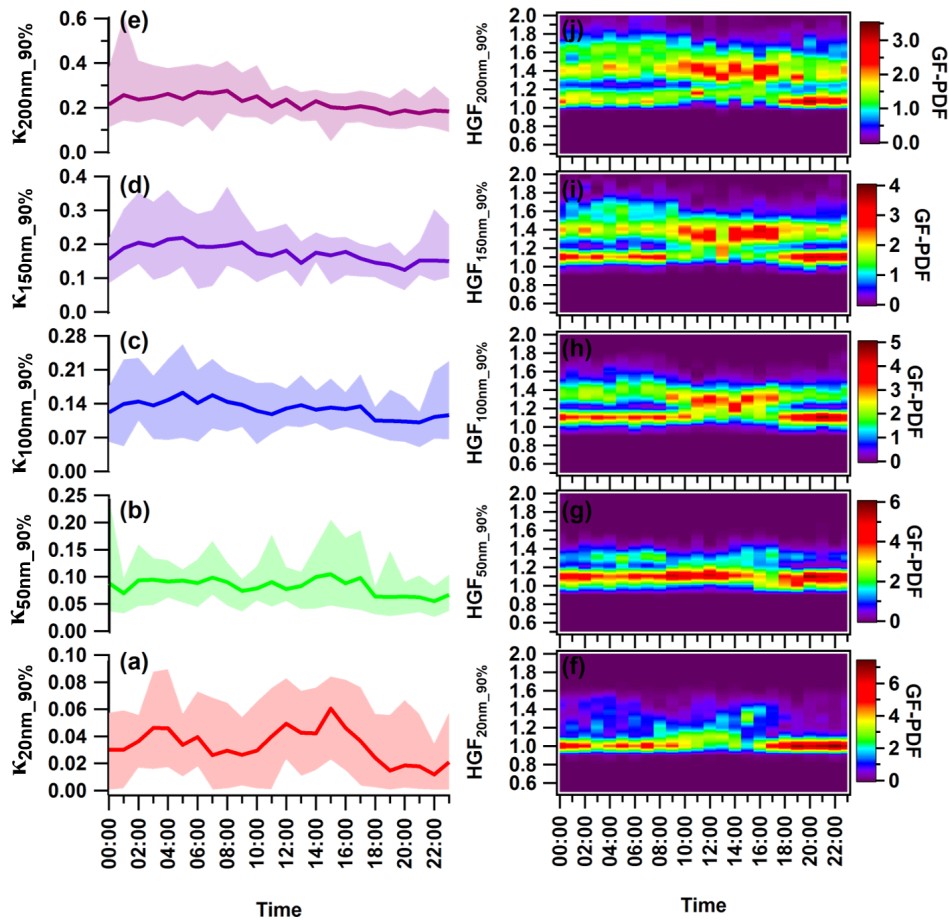




**Figure 4: Diurnal variance in the hygroscopic parameter kappa ($\kappa$) of nucleation mode particles (a) 20 nm ($\kappa_{20nm\_90\%}$),**
**Aitken mode particles (b) 50 nm ($\kappa_{50nm\_90\%}$), and Accumulation mode particles (c) 100 nm ($\kappa_{100nm\_90\%}$), (d) 150 nm**
**($\kappa_{150nm\_90\%}$), and (e) 200 nm ($\kappa_{200nm\_90\%}$) and hygroscopic growth factor of (f) 20 nm (HGF$_{20nm\_90\%}$), (g) 50 nm**
**(HGF$_{50nm\_90\%}$), (h) 100 nm (HGF$_{200nm\_90\%}$), (i) 150 nm (HGF$_{150nm\_90\%}$), and 200 nm ($\kappa_{200nm\_90\%}$) aerosol particles. The**
**solid line represents diurnal average values, and the upper and lower shaded area represents 95 and 5 percentile values**
**of corresponding average values. Different color coding has been used to represent various size-specific kappa values.**
**The color scale represents the growth factor probability density function of hygroscopic growth factor.**
**3.2.3 Driving Factor of Hygroscopicity**
A correlation analysis was carried out between measured chemical species and aerosol to explore the factors
governing aerosol hygroscopicity, as shown in the figure 5. Organic aerosol was observed negatively impact $\kappa$,
explained by a negative correlation (figure S7(a)). This negative correlation of OA with $\kappa$ is also observed in India
(Bhattu et al., 2016; Mandariya et al., 2020b) and worldwide (Enroth et al., 2018; Hong et al., 2014; Kawana et
al., 2016; Kitamori et al., 2009; Wang et al., 2018a; Wu et al., 2013a). This result indicates that primary
constituents dominated the OA during high loading, considered nearly hydrophobic or less hygroscopic. In
addition, the current study observed that an enhancement of 10% of OA by volume in 200 nm aerosol particles
would be responsible for a 4% decrement in its hygroscopicity (figure S7(a)). Interestingly, ammonium sulfate
and nitrate showed a positive but poor correlation with hygroscopicity. It could be due to sulfate and nitrate aerosol
dominating the bigger particles (>200nm). However, a 10% enhancement of AS by volume was found to be
responsible for the enhancement of hygroscopicity only by 1.6%. Besides, figure 5(a) shown an increasing volume
fraction of ACl in PM$_1$ with an increase in aerosol hygroscopicity, and this strong positive correlation is
responsible for an enhancement in kappa by 4.2% over the increment of 10% ACl by volume, which was the
highest among all chemical species. Further, ammonium chloride has a more significant water uptake potential
(Chen et al., 2022; Zhao et al., 2020), which can be justified by the solid correlation of ALWC with a mass fraction
of ACl in PM$_1$ as shown in figure 5(b). This indicates that particles with a more considerable ammonium chloride
fraction uptake more water vapor, leading to higher hygroscopic aerosol particles. It is clear that the increases in
ammonium chloride fraction enhanced aerosol liquid water content and led to higher hygroscopicity of aerosol
particles. A Recent study in Delhi by Chen et al. (2022) unveils that ammonium chloride fraction in PM$_1$ aerosol
enormously enhances during the higher relative humidity conditions during the winter season due to the co-
condensation of semivolatile ammonium chloride with water vapor on the particles and leads to enhance water
uptake and lead severe winter haze in Delhi.  The very high volume fractions (>30%) of ACl in atmospheric NR-



PM$_1$ were observed episodic, suggesting a high fraction of Cl in the particle phase is strongly dependent on excess
ammonia in the atmosphere. These results indicate that ammonia is the controlling factor for chloride partitioning
in the particle phase, resulting in high aerosol water content under high RH and lower temperature conditions. As
the ACl is strongly dependent on the RH and temperature.

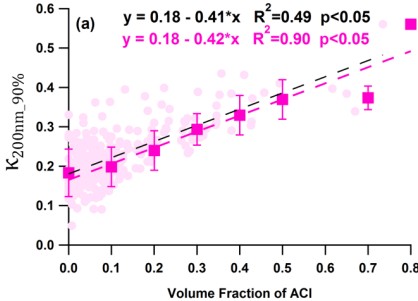 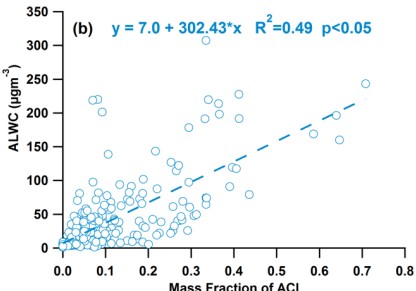


**Figure 5: Correlation plot for (a) κ$_{200nm\_90\%}$ vs volume fraction of ammonium chloride aerosol (VF$_{ACl}$) and (b) aerosol**
**liquid water content (ALWC) vs mass fraction of ammonium chloride (MF$_{ACl}$).**
**3.2.4 Hygroscopicity during high biomass burning (H-BB), high-hydrocarbon like OA (H-HOA), high-Cl**
**(H-Cl), and Clean Periods**
Delhi's atmosphere is a complex array of chloride and organic aerosol sources like combustion (crop residue,
agriculture waste, medical waste, municipal waste, plastic, etc., burning) and industrial sources. Therefore, all
episodic events were classified into three to investigate the impact of chloride and OA on aerosol hygroscopicity.
First, high biomass burning (H-BB) event; second, high-hydrocarbon like OA (H-HOA) event; and third, high-Cl
(H-Cl) event according to the substantial fractional contribution of their markers in respective periods. In addition,
we also classified a "Clean Period" where PM$_1$ loading was less than 25 percentile ($\leq$ 38.7 µgm$^{-3}$) of the sampling
period. Further, aerosol chemical composition data were filtered according to hygroscopic parameter data for
further analysis. By performing so, data information that is characteristic of the local emission and atmospheric
chemistry in question and the effects of various potential transported air mass types can be retrieved. It is valuable
to extract any possible information about aerosol sources and transformation process evaluation to interpret its
influence on the aerosol's hygroscopicity.
**3.2.4.1 Relatively Clean Period**





The relatively Clean period was predominantly dominated by E, S-E winds; however, pollution was associated
with calm winds, as illustrated in figure S9. All BBOA, HOA, and ACl were observed to be associated with
similar sources and found an excellent association with ambient relative humidity. The mean concentration of
organic aerosol, ACl, AN, and AS was observed at $11.0 \pm 6.4$, $1.4 \pm 1.1$, $3.0 \pm 1.5$, and $4.4 \pm 2.2$ $\mu gm^{-3}$,
respectively. These mass concentrations were significantly lower than in other specified periods. However, OA
was still the dominant species, with 56% by volume in the $PM_1$, as indicated in figure 8. Among all OA factors,
HOA was predominantly dominated in OA with 33%, although secondary organic aerosol confined the overall
54.4% of OA. Secondary OA is defined with relatively higher oxidized OA, and the oxidation state of OA
positively impacts OA hygroscopicity (Kim et al., 2017; Richard et al., 2011; Wu et al., 2013a). The Clean period's
mean hygroscopicity of 20, 50, 100, 150, and 200 nm particles were observed at $0.03 \pm 0.02$, $0.09 \pm 0.04$, $0.14 \pm$
$0.06$, $0.22 \pm 0.09$, and $0.27 \pm 0.07$, respectively, significantly ($p<0.05$) different to each other. However, the
accumulation particle's (200 nm) hygroscopicity was not significantly ($p>0.05$) higher than the 150 nm particles.
The hygroscopicity increment with size from 20 to 200 nm can also be explained by the fractional increment of
more hygroscopic (GF>1.2) particles relative to nearly hydrophobic or less hygroscopic particles (GF<1.2).
Nucleation particles, 20 nm was dominated mostly by less hygroscopic particles ($76.8 \pm 21.7\%$), indicates
influence by fresh emission sources, whereas, Aitken (50 nm) and Accumulation (200 nm) size aerosol were
confined with $69.3 \pm 14.7$ and $25.4 \pm 10.8$ less hygroscopic particles, respectively. These results point out that
accumulation-size aerosols dominated secondary aerosols, which can also supports their GF-PDF as shown in
figure 7(a). Nucleation size aerosol particles (20nm) showed nearly mono modal GF-PDF with the mode of unit
growth factor. In contrast, the mode shifted towards the higher end as aerosol size increased and GF-PDF shifted
from unit to multi-mode.
**3.2.4.2 High-Cl (H-Cl) events**
H-Cl events representing the substantial loading of ACl on the receptor site, as shown in the pie chart in figure 8,
were chosen mainly due to the significant jump (>20%) in fractional volume contribution $NH_4Cl$ ($\varepsilon_{ACl}$) in the $PM_1$
aerosol. This period observed apparent surface wind from W-direction, although WNW, WSW, and SE winds
also influence the site, as shown in figure 6(b). BBOA and HOA are potentially contributed from the WNW and
SE directions, as explained in the bipolar plot figure 6(c & d), and seem to come from a similar local source.
Among inorganic species, ACl observed excellent association with ambient RH, as shown in figure 6(e) and (f),
indicating the atmospheric gaseous HCl neutralized with $NH_3$ gas in the presence of atmospheric water content.
HCl sources could be coal power plants, trash burnings in solid waste dumping sites, and other industries located



in the W-WSW direction (Gani et al., 2019), as shown in the map in figure 6(a). Atmospheric high Cl events are
potentially dominated by trash burning in Delhi during winter (Shukla et al., 2021; Tobler et al., 2020). Moreover,
bipolar plots (figure 6(e and f)) suggest that ACl formation under high RH conditions associated with a relatively
calm atmosphere trigger particles' hygroscopicity. This hypothesis can be supported with a good association of
aerosol liquid water content (ALWC) as discussed in previous section. Furthermore, GF-PDF of all size particle
marked relatively more fractionaly contribution of secondary mode particles as showed in figure 7(d). Overall
more hygroscopic (HGF90%>1.2) particles were marked by 42, 47, 50, 74, and 83% contributions in the 20, 50,
100, 150, and 200 nm size particles, respectively. Hence, ACl is a critical factor to enhance aerosol hygroscopicity
to trigger fog/haze formation under higher RH and colder atmospheric conditions as discussed in the previous
section.
Similarly, Gunthe et al. (2021) observed that high local emission of hydrochloric acid in Delhi during February-
March gets partitions into aerosol liquid water under high humid conditions, enhancing the water uptake capacity
of aerosol sustain particle's hygroscopic growth, result in fog/haze formation. Moreover, worldwide studies on
size-resolved hygroscopicity observed Cl less than 1%, so they omitted ACl as an aerosol constituent into the
discussion. In addition, the current study did not find any strong correlation of $\kappa$ with AS and AN. It could be due
to their association with larger particle sizes. Besides, ACl could be associated with comparatively lower size
particles ($\leq$ 200 nm). Furthermore, in context to look influence of air mass trajectories, we further mapped
aerosol's constituents in the association of air mass back trajectories in PSCF to see the potential area source
contribution influencing the aerosol evaluation processes, ultimately aerosol's hygroscopicity. However, we did
not find any back trajectory influencing the receptor site, as all trajectory endpoints were observed above the
planetary boundary layer height.

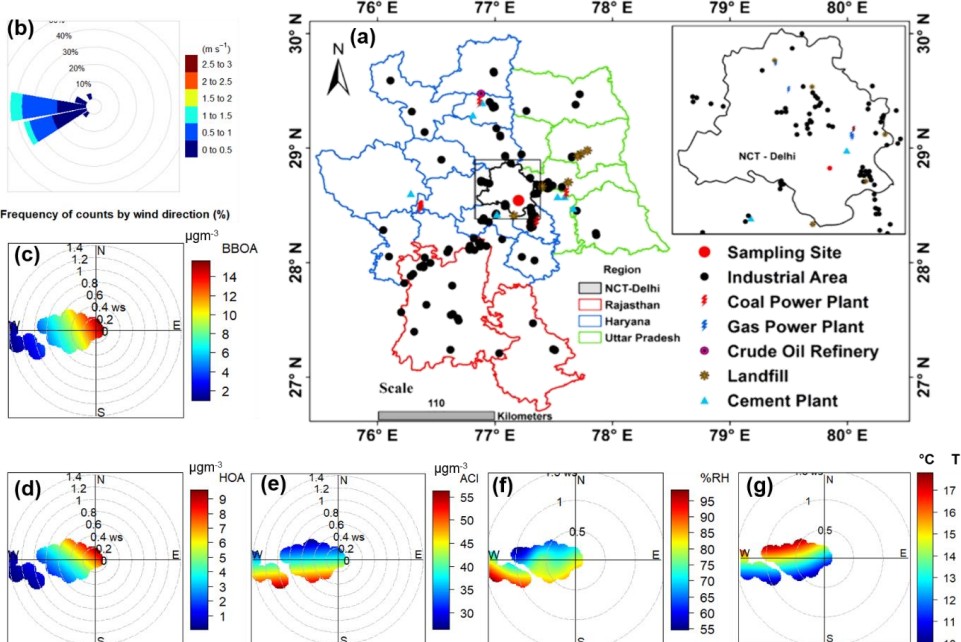

**Figure 6: Map of (a) Delhi showing various types of industries located in the region and nearby locations, (b) wind rose diagram and conditional bi-polar plots showing variation in mass concentration of (c) biomass burning OA (BBOA), (d) hydrocarbon like OA (HOA), (e) ammonium chloride (ACl), (f) % ambient relative humidity (RH), and (g) ambient temperature (T), with wind direction (WD) and wind speed (WS) during H-Cl events. A background map showing various industrial locations was adapted from Rai et al. (2020).**

### 3.2.4.3 High biomass burning (H-BB) Events

High BB periods were noted during the initial period (1-12 February) of the field campaign. The predominant surface wind circulations were from W, W-WNW, and W-WSW directions (figure S8(b)). The aerosol was dominated by local emissions, as aerosol constituents are mainly associated with slower wind circulations from landfill sites, industrial areas, and coal power plants, as shown in figure S8(a). Further, it could justify the potential source contribution function (PSCF) analysis considering 48-hr air mass back trajectories, as shown in figure S10. Therefore, BBOA possibly contributed from the open local biomass burning activities at landfill sites or others. Biomass burning organic aerosol confined the most considerable fraction, 39%, of organic aerosol, following HOA, 28%. Figure S5(b, c, and d) clearly shows that BBOA and HOA have similar source profiles but differ from the ACl source. Moreover, ACl was not found to have a good association with ambient RH and but was associated with nearby coal power plant's emissions. However, 48 h air mass back trajectories indicated that the current city





was also influenced by air mass from some parts of Uttar Pradesh, Punjab, and Haryana. These states are the
potential hub of crop residue burning, industrial activities and brick kilns. These cities have a substantial fraction
of OA in $PM_1$ and OA mainly affected by biomass activities during winter. The H-BB event captured a
considerable volume fraction, 71% of OA in $PM_1$ and BBOA contributed almost 39%, as illustrated in the figure
8. So, lower inorganic to OA ratio was a potential factor in decreasing the aerosol hygroscopicity in H-BB events.
Further, a primary organic aerosol contribution was enhanced during this event and, on average, raised to 67%.
OA loading inversely affects the aerosol's hygroscopicity. Mandariya et al. (2020) reported a similar observation
in Kanpur, and the authors suggested that the contribution of primary biomass burning (BBOA) and hydrocarbon-
like OA adversely affects aerosol hygroscopicity. BBOA showed a good negative correlation with the
hygroscopicity of 200 nm particles, supporting the following conclusion. Apart from this, the Nucleation size
particle (20 nm) showed $0.02 \pm 0.02$ hygroscopicity parameter with mono mode GF-PDF with the unit mode
(figure 7(b)) and confined $83.7 \pm 18.6$ % nearly hydrophobic particles. Furthermore, as aerosol size increased,
hygroscopicity parameter ($\kappa_{H\text{-}TDMA\_90\%}$) enhanced significantly ($p<0.05$) as the contribution of relatively
secondary aerosol particles (GF>1.2) increased with aerosol size. Accumulation size aerosol, 100 nm contributed
approximately 54% by nearly hydrophobic (GF<1.2) and 46% by more hygroscopic (GF>1.2) particles.

### 3.2.4.4 High-HOA (H-HOA) Events

H-HOA events were identified based on the considerable mass concentration and fraction of HOA in the organic
aerosol. These periods were noted generally 19:00 hr to Morning 09:00 hr during 22-23 and 26-27 February and
4, and 7-8 March as indicated in figure 1. PSCF explore the probability of impacts of long-range transported
aerosol. Interestingly, it was observed that air masses over Delhi, Haryana, and Uttar Pradesh were potentially
associated with hydrocarbon-like OA (figure S10). BBOA also followed a similar path as HOA. However, the
potential area source of ACl was the nearby region of Delhi and Haryana. The HOA loading was significantly
($p<0.05$) higher than in H-BB, H-Cl and Clean periods. However, emission sources were different during both H-
HOA and H-BB periods. As HOA was the potential contributor to OA, it is likely the critical constituent to
modulate aerosol hygroscopicity in the region during these events. HOA is mainly considered hydrophobic
(Duplissy et al., 2011). Therefore, elevated HOA contribution (41%) in OA could be responsible for lower $\kappa$ in
these events. The overall hygroscopicity of 20, 50, 100, 150, and 200 nm size particles was recorded as $0.01 \pm$
$0.01, 0.06 \pm 0.03, 0.11 \pm 0.03, 0.14 \pm 0.04$, and $0.17 \pm 0.05$, respectively. The predominant fractional contribution
of primary aerosol particles (GF<1.2) seems to be a reason for this lower hygroscopicity of particles, as shown in
figure 7(c). Overall, OA predominantly constitutes the fraction in the $PM_1$, and primary OA contributed





approximately 60% in OA. However, relative increment in the contribution of other more hygroscopic constituents
like secondary organic aerosol (LO-OOA and MO-OOA), ACl and ammonium sulfate (AS) in the aerosol possible
tried to balance the negative impact high-HOA on which limited $\kappa$.

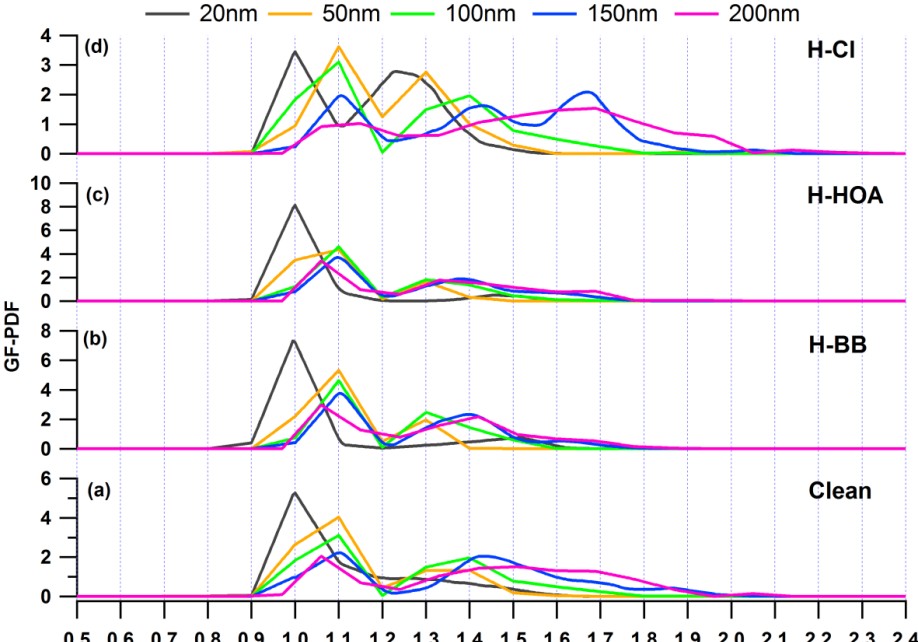


**Figure 7: Growth Factor Probability Density Function (GF-PDF) of 20, 50, 100, 150, and 200 nm aerosol particles for**
**the (a) clean, (b) H-BB, (c) H-HOA, and (d) H-Cl periods.**
**3.2.4.4 Comparison of $\kappa$ of different events**
We choose a 200 nm accumulation particle size particle representing the bulk aerosol chemical composition to
compare the aerosol hygroscopicity among various periods. Further, in the present study, the mode of particle-
volume size distribution varied from 400 nm to 600 nm particle dry mobility diameter. Therefore, 200 nm size
accumulation particles are the best choice to compare hygroscopicity parameters among different periods
considering bulk aerosol composition in various mentioned periods. In addition, a good Pearson's r value, 0.76,
was found among $\kappa_{200nm\_90\%}$ and $\kappa_{chem\_90\%}$, derived from the dry $PM_1$ particle's chemical composition measured
from the ACSM based on the ZSR mixing rule (Stokes and Robinson, 1966), which justifies our choice.



The H-Cl event noted the highest value (0.36 ± 0.06) of $\kappa_{200nm\_90\%}$ against H-BB (0.18 ± 0.04), H-HOA (0.17 ±
0.05), and Clean (0.27 ± 0.07) events, as illustrated in figure 8. The H-Cl event observed that the average $\kappa_{200nm\_90\%}$
value was significantly ($p<0.05$) higher than those observed in other events. It means that a substantial increment
in Cl emission in the Delhi region could significantly enhance the aerosol liquid water content leading to higher
aerosol hygroscopicity, which can further strengthen cloud condensation nuclei formation, possibly triggering
haze/fog events in Delhi NCR (Gunthe et al., 2021). These results suggest that controlling the open trash/waste
burning in the region could help control Cl emission, which leads to minimizing the haze/fog formation possibility
during high atmospheric conditions. However, the difference in $\kappa_{200nm\_90\%}$ values between H-BB and H-HOA
events was not observed significantly ($p>0.05$), possibly due to the relative changes in primary, secondary OA,
and inorganic species. In the H-HOA events, the negative effect of a significantly higher fractional (41%)
contribution of HOA to OA possibly balances with a positive impact of a 7% increment in secondary OA relative
to H-BB. Worldwide studies (Jimenez et al., 2009; Mandariya et al., 2019; Sun et al., 2013) reported secondary
organic aerosol associated with a higher O/C ratio, and several studies reported that the O/C ratio positively
correlated to $\kappa$ (Jimenez et al., 2009; Kim et al., 2020) as described in the earlier text. Furthermore, impacts of
5% decrement in ACl during H-HOA event concerning H-BB event possibly managed by 7% increment in AS
fraction. Overall, these relative changes in aerosol constituents worked to insignificant changes in $\kappa$ during H-BB
and H-HOA periods. Nevertheless, H-BB and H-HOA events witnessed significant ($p<0.05$) lower hygroscopicity
compared to a relatively cleaner atmosphere. The aerosol associated with relatively cleaner events was with a
higher inorganic-to-organic ratio. In addition, the aerosol in clean periods comprised a significantly higher fraction
of secondary organic aerosol, which could be the reason for the higher hygroscopicity associated with organic
aerosol compared to other events. Worldwide (Aiken et al., 2008; Cerully et al., 2015b; Chakraborty et al., 2016b;
Mandariya et al., 2019) studies have reported that organic aerosol loading inversely impacts the oxidation/aging
process of OA. Overall, all these were responsible for higher hygroscopicity in relatively cleaner periods.



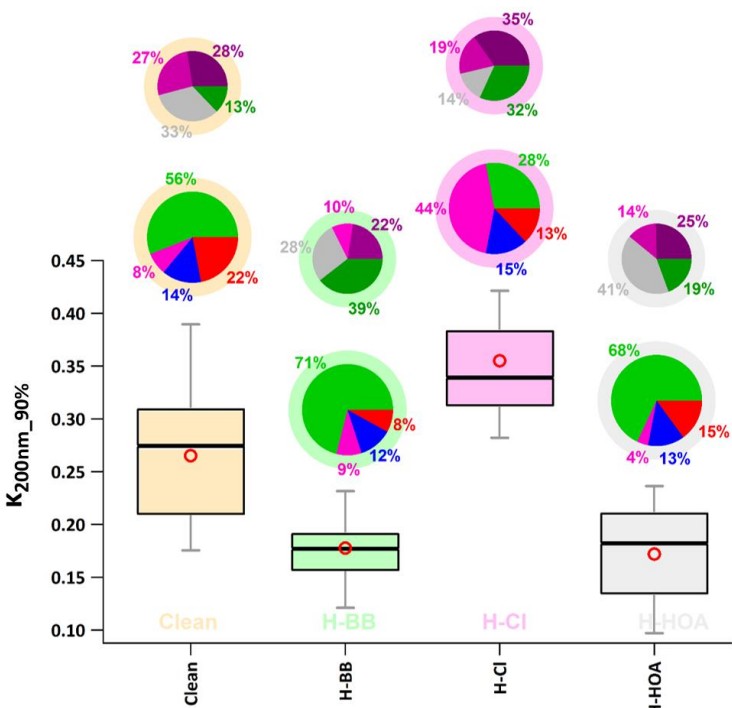


**Figure 8: Box plot showing variation in H-TDMA measured hygroscopic parameter of 200 nm size particles $\kappa_{H\text{-}TDMA}$**

**($\kappa_{200nm\_90\%}$) in high biomass burning (H-BB), high-chloride (H-Cl), and high-hydrocarbon like organic aerosol (H-**

**HOA) events. Different colors represent respective events in the plot. A bigger pie chart represents the overall average**

**volume fractional contribution of various aerosol species indicated by color-coding. In addition, minor pie charts**

**described the event average mass fractional contribution of different OA species in OA. Diffused ring color of the pie**

**chart displays the respective event.**

**4. Conclusions**

The current study explored the temporal variation of Nucleation (20 nm), Aitken (50 nm), and Accumulation (150

and 200 nm) mode particle hygroscopicity in Delhi during the winter period, February-March 2020. In addition,

the present study highlighted the hygroscopicity variation in relatively higher chloride, biomass burning, and

hydrocarbon-like organic aerosol. Because, Delhi highlighted as one of the most polluted cities, faces high

chloride pollution episodes during winter haze/fog. Therefore, we reported temporal variation in the size-resolved

hygroscopic parameter ($\kappa_{H\text{-}TDMA\_90\%}$) at the sub-saturated level (90% RH) for the first time in Delhi. However, we

reported hygroscopicity of nucleation and Aitken mode particles for the first time in India.



The observed $\kappa_{H\text{-}TDMA\_90\%}$ ranged from .00 to 0.11 (0.03 ± 0.02), 0.05 to 0.22 (0.11 ± 0.03), 0.05 to 0.30 (0.14 ±
0.04), 0.05 to 0.41 (0.18 ± 0.06), and 0.05 to 0.56 (0.22 ± 0.07) for 20, 50, 100, 150, and 200 nm aerosol particles,
respectively. The study period's mean value of the hygroscopicity parameter was significantly ($p<0.05$) enhanced
with the size of the particles. $\kappa_{20nm\_90\%}$ and $\kappa_{50nm\_90\%}$ were observed to show dynamic diurnal variation. In contrast,
as particle size increased in accumulation mode particles, noontime variation became flatten, which found an
attribute with neutralizing positive and negative impacts of increment and decrement in volume fraction of $NH_4Cl$
and OA, respectively. Interestingly, the variation in $\kappa_{200nm\_90\%}$ was observed potentially due to variation in $NH_4Cl$
and OA instead $(NH_4)_2SO_4$.
Furthermore, the pollution episodes were generally associated with local biomass burning and industrial and
waste-burning emissions in Delhi and nearby regions. We mainly focused on emphasizing the impacts of high
biomass burning (H-BB), high hydrocarbons like OA (H-HOA), and high chloride emission (H-Cl) on aerosol
hygroscopicity and their relative comparison with the cleaner period. The H-Cl period was observed significantly
($p<0.05$) higher hygroscopicity (0.35 ± 0.06) compared to H-BB (0.18 ± 0.04), H-HOA (0.17 ± 0.05), and
relatively cleaner period (0.27 ± 0.07). However, H-BB and H-HOA showed no significant difference in
hygroscopicity. However, they witnessed lower aerosol hygroscopicity concerning a relatively cleaner
atmosphere. It could be attributed to the lower organic aerosol loading and higher inorganic-to-organic aerosol
ratio in the aerosol. High atmospheric chloride aerosol (ammonium chloride) was observed most affectionately to
the atmospheric water, leading to higher aerosol liquid water content at high chloride events. In addition, a high
volume fraction of ammonium chloride in aerosol enhanced the aerosol hygroscopic nature. Furthermore, our
results based on episodic interpretation revealed that chloride emission is a significant concern in Delhi, which
enhances the aerosol's hygroscopicity, helps to act as CCN to form fog droplets during winter colder days, and
leads to fog/haze formation in Delhi. In addition, high chloride present in aerosol overcomes the negative impact
of high OA loading on CCN activity. Hence, our results suggest controlling waste materials' open burning to
decrease the haze/fog events in Delhi during winter.
**Supporting Information**
Supplementary pieces of information are mentioned in the supplementary file.
**Data availability.** Data can be accessed at the following repository:
https://web.iitd.ac.in/~gazala/publications.html (Mandariya et al., 2023).



**Author contributions**

AH, MMVH, NAB, and GH operated aerosol instrumentation and collection of data on-board in Delhi. KP

analysed the ACSM data. AKM, AH, and GH conceptualized the structure of the manuscript. AKM analysed,

evaluated H-TDMA data, and wrote the manuscript. AH analysed MPSS data. AKM, AH, KP, JSA, LHR, AW,

and GH internally reviewed the manuscript and helped to write the manuscript.

**Corresponding Author**

Gazala Habib (gazalahabib@civil.iitd.ac.in) and Alfred Wiedensohler (ali@tropos.de)

**Competing interests**

The authors declare that they have no conflict of interest.

**Acknowledgment**

The authors thankful to Dr. Martin Gysel, Aerosol Physics Group, Paul Scherrer Institute, Switzerland, for

providing TDMAinv toolkit for HTDMA data correction.

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
