# Peer review of "Measurement report: Hygroscopicity of Size-Selective Aerosol"

_EGUsphere, 2023_

## Referee Comment (RC1)

Review to "Hygroscopicity of Size-Selective Aerosol Particles at Heavily Polluted Urban Atmosphere of Delhi: Impacts of Chloride Aerosol"

The authors present field measurements of size-resolved aerosol hygroscopic growth at 90% RH and bulk aerosol composition of non-refractory $PM_1$ during wintertime in Delhi, India, and investigate the impacts of chloride on aerosol hygroscopicity and its potential to enhance aerosol-bound liquid water. The paper provides observational evidence of Ammonium Chloride as the major contributor to aerosol hygroscopic growth and liquid water content in Delhi, which highlights the role of Ammonium Chloride in aerosol-water interaction and related haze development. I would recommend publication once the following concerns are addressed.

**Major comments:**

1) The manuscript is a bit long and wordy to me. The authors put too much effort on the overview of the 1.5-month field measurement, and enumerate the ranges of many aerosol properties, e.g., $PM_1$ mass concentration, chemical composition mass of different species. For example, "BBOA mass concentration varied between 0.0 to 134.7 $\mu g/m^3$", I feel sentences like this are not as informative, and should be reduced as much as possible.

2) I strongly suggest the authors add a representative case study including major gaseous pollutants, aerosol size distribution, chemical composition, and GF-PDF of 1~2 sizes, to showcase the driving effect of $NH_4Cl$ on aerosol hygroscopicity and see if $NH_4Cl$ exists in all size ranges (i.e., 20~200, from GF-PDF).

3) As shown in Fig. 3a, the $\kappa$ of 20 nm particles look quite scattered to me. For the GF-PDF of 20 nm, I am curious about how many counts of 20 nm particles were sampled for each cycle. As the counting statistics may affect the inversion of GF-PDF. According to a recent study, a total of at least 100 particle counts might be a requirement for reliable GF-PDF inversions (https://doi.org/10.5194/amt-15-2579-2022).

4) As shown in Fig. 4, the diurnal variation of $\kappa$ is overshadowed by the hydrophobic mode (e.g., HGF<1.2 for 100 nm). I would suggest the authors to isolate the hygroscopic mode and calculate the corresponding $\kappa$. To do so, you could either set a fixed threshold of HGF or fit the bi-modal GF-PDFs and calculate $\kappa$ using the more hygroscopic mode. By doing this, the authors could probably compare the $\kappa$ of the hygroscopic mode to pure $NH_4Cl$.

5) Line 351-352: The authors attribute the two-peaked pattern in the GF-PDF to daytime photochemical reactions. If that is the case, why does the HGF decrease at noontime when photochemical activities are supposed to be even stronger.

6) Regarding the minor difference in $\kappa_{200nm\_90\%}$ for H-BB and H-HOA events, I doubt if that because the two events are not well separated from each other. Do the authors have a general criterion for

separating the three different events, or at least show how much overlapping there is between the two events.

**Minor comments:**

1) Line 101: Use subscript in "(NH4)2SO4".

2) Line 159: Wrong expression in equation (1).

3) Line 236: Full spell "MPSS" where it is mentioned for the first time.

4) Line 239-241: The value of OA mass concentrations does not seem to be consistent with that reported in Gani et al., (2019).

5) Line 300: $HGF_{90\%\_200nm}$ of "1.12-1.179", but average to $1.41 \pm 0.09$?

6) Line 405: Full spell "ALWC" where it is mentioned for the first time.

**Comments for figures:**

1) Figure 1a and 1b: The lines are overlapped with the shaded boxes, looks like in-continuous data. Describe the shaded boxes of different colors in the caption.

2) Figure 2o: The y-axis label is blocked.

3) Figure 2d: The diurnal WD pattern looks quite different from the wind rose plot (i.e., Figure S6). The latter suggested a negligible fraction of southerly wind.

4) Figure 3: line 332, is Fig. 3d for 150 nm?

5) Figure S5: Use a consistent unit, ppb or $\mu g\ m^{-3}$.

6) Figure 8: Add a legend for the pie chart here.

---

## Referee Comment (RC2)

The manuscript " Measurement report: Hygroscopicity of Size-Selective Aerosol Particles at Heavily Polluted Urban Atmosphere of Delhi: Impacts of Chloride Aerosol" revealed the wintertime chloride emission in the Delhi region governing the enhancement of aerosol hygroscopicity and aerosol-bound liquid water that trigger Delhi's fog episodes. The manuscript is written well and within the interest of the scientific communities. However, there are many gaps in the quality of presentation and lack of clarity in the manuscript.

**Major Comments**

- The author did not present the schematic of the experimental design. Therefore, it is difficult to understand the different instruments used in the study.

- There is a lack of clarity on the classification of different events e.g., H-BB, H-HOA, H-Cl and clean. For example, how was the event classification made based on the aerosol chemical compositions? There is missing information about these events in figure 1 caption. It is recommended that the author should add a table to the text to discuss the event classification explicitly.

- The mathematical equations used in the text should be cross verified.

**Minor comments**

Page 1 and Line 27: Expand HTDMA

Page 1 and Line 33: Expand OA

Page 3 and Line 67: Expand IGP

Page 5 and Line 137: This is a repeated sentence.

Page 6 and Line 159: The equation is not correct.

Page 6 and Line 169: It is not clear the modified ion pairing scheme: what is the difference between SA and AS

Page 8 and Line 195: The author should discuss the source of the gas and meteorological data. At what height the met parameters were measured?

Page 8 and Line 196: The author talked about PNSD. It is not clear how they measured it? Is it from the HTDMA or additionally a size spectrometer was used. A detailed schematic experimental design is needed.

Page 8 and Line 205-207: Reference is missing.

Page 8 and Line 213: It is not clear how the intensity of biomass burning activities was determined.

Page 9 and Line 220: Author should explain the nighttime peak of SO2.

Page 9 and Line 235: It is not clear about MPSS. Is it a separate instrument associated with the experimental design? If so, why was the MPSS data not presented in this study?

Page 9 and Line 240: … average mass concentration 46.5 ±39.6 ug/m3 consistent with 112 ug/m3…. This is not clear.

Page 12: The y- axis of diel Cl plot is not clear.

Page 17 and Line No.398: Author should explain why two linear regressions are drawn in the correlation plot (example Fig. 5a).

Page 18 and Line No.418: Author should provide the ALWC vs mass fraction of AN and AS in the supplement.

Page 19 and Line No 434. The author should clearly mention the dates they consider for a relatively clean period.

Page 22 and Line No 505-507. Is it 39% of BBOA by mass? Figure 8 is not clear. The color coding should be clarified in the plot.

Page 22 and Line No 505-507. The dates and times of the event should be clarified in the figure 1 caption.

Page 23 and Line 535: The x-axis label is missing.

Page 25 and Line 583: However,…time in India…This statement is not true.

Supplements

Page 2 and Line 23: Author should present the time series data of MPSS during the study period.

Page 7 and Line 80: I don't see any difference in the probability distributions of BBOA, HOA and ACL. The Author should clarify it.

---

## Author Comment (AC1)

**Response to the Referee (RC) 2**

The manuscript "Measurement report: Hygroscopicity of Size-Selective Aerosol Particles at Heavily Polluted Urban Atmosphere of Delhi: Impacts of Chloride Aerosol" revealed the wintertime chloride emission in the Delhi region governing the enhancement of aerosol hygroscopicity and aerosol-bound liquid water that trigger Delhi's fog episodes. The manuscript is written well and within the interest of the scientific communities. However, there are many gaps in the quality of presentation and lack of clarity in the manuscript.

Major Comments

- The author did not present the schematic of the experimental design. Therefore, it is difficult to understand the different instruments used in the study.

Response:

Thank you for your constructive comments. Your suggestion seems very legitimate. As suggested, this comment has been addressed in the revised manuscript. The schematic diagram of sampling instruments was added in the revised manuscript.

*(Line 83-88 and 119-121) "Real-time atmospheric aerosol measurements were conducted simultaneously using Hygroscopic-Tandem Differential Mobility Analyzer (H-TDMA), Mobility Particle Size Spectrometer (MPSS), and Aerodyne Aerosol Chemical Speciation Monitor (ACSM, Aerodyne Research, Billerica MA) during winter ($1^{st}$ February 2020 to $16^{th}$ March 2020) at the Indian Institute of Technology (IIT) Delhi in Block 5, at the height of nearly 15 m as shown in Fig. 1. The lab-2 is situated at the height of 15 m above the ground level and lab-1 is 50 m apart from lab-1."*

[Figure]

*Figure 1: Schematic diagram of the inlet systems for aerosol sampling instruments. The blue and red sampling lines indicate the ambient air and dehumidified (RH<25%) ambient air, respectively.*

- There is a lack of clarity on the classification of different events e.g., H-BB, H-HOA, H-Cl and clean. For example, how was the event classification made based on the aerosol chemical compositions? There is missing information about these events in figure 1 caption. It is recommended that the author should add a table to the text to discuss the event classification explicitly.

Response:

We sincerely thank the reviewer for pointing it out. Instead of a table, we add statements explaining these events' classification.

*(Line 170-175) "Furthermore, based on the significant mass concentration peaks of BBOA, HOA, and Cl in the temporal variation, respectively, three different events were characterized: 1) High-residential or biomass burning (H-BB), 2) High-hydrocarbon-like OA (H-HOA), and 3) High-chloride (H-Cl) period. In addition, the "Clean Period" was defined where PM₁ loading was less than 25 percentiles (≤ 38.7 µgm⁻³) of the sampling period. The starting and end time of the event was defined by the starting the increment in the concentration and reaching the starting value while the concentration decreased."*

- The mathematical equations used in the text should be cross verified.

Response:

Thank you for your correction. We modified the mathematical equations used in the text in the revised manuscript

*(Line 182-199)*

**Case-1** $R_{SO_4}(NH_4 \ to \ SO_4) \leq 1$

$SA = 98.0795 \times max(0, (n_S - n_A))$

$ABS = 115.11 \times n_A$

$AS = 0$

$AN = 0$

$ACl = 0$

**Case-2** $1 < R_{SO_4} < 2$

$SA = 0$

$$ABS = 115.11 \times \left((2 \times n_S) - n_A\right)$$

$$AS = 132.1405 \times (n_S - n_A)$$

$$AN = 0$$

$$ACl = 0$$

**Case-3 $R_{SO_4} \geq 2$**

$$SA = 0$$

$$ABS = 0$$

$$AS = 132.1405 \times n_S$$

$$AN = \left(min\left(\left(n_A - \left(\frac{ABS}{115.11}\right) - \left(\frac{(2 \times AS)}{132.1405}\right)\right), n_N\right)\right) \times 80.0434$$

$$ACl = \left(min\left(n_C, \left(n_A - \left(\frac{ABS}{115.11}\right) - \left(\frac{2 \times AS}{132.1405}\right) - \left(\frac{AN}{80.0434}\right)\right)\right)\right) \times 53.54$$

Minor comments

Page 1 and Line 27: Expand HTDMA

Response:

As suggested, this comments have been incorporated in the revised manuscript.

*(Line 18-22) "In this study, we present the measurement results of bulk aerosol composition of non-refractory PM₁ from ACSM and size-resolved (Nucleation, Aitken, and Accumulated mode particles) hygroscopic growth factor and associated hygroscopicity parameter at 90% relative humidity (RH) measured using H-TDMA (Hygroscopic-Tandem Differential Mobility Analyser) at Delhi Aerosol Supersite (DAS) for the first time."*

Page 1 and Line 33: Expand OA

Response:

As suggested, this suggestion has been incorporated in the revised manuscript.

*(Line 31-32) "Additionally, the high chloride content in aerosols appears to counteract the negative effects of high organic aerosol (OA) levels on cloud condensation nuclei (CCN) activity."*

Page 3 and Line 67: Expand IGP

Response:

Thank you for your correction. We modified the sentence in the revised manuscript.

*(Line 64-65) "In past decades, fast economic growth and industrialization in the Indo Gangetic Plain (IGP) led to severe air quality during wintertime (Wester et al., 2019)."*

Page 5 and Line 137: This is a repeated sentence.

Response:

The text '*The humidity sensor of DMA2 was automatically calibrated with 100 nm ammonium sulfate particles after each scan cycle.*' was removed as suggested.

Page 6 and Line 159: The equation is not correct.

Response: We sincerely thank the reviewer for pointing it out. As suggested, the equation (1) has been corrected in the revised manuscript.

*(Line 152)*

$$\kappa_{H-TDMA\_90\%} = (HGF\_90\%^3 - 1)\left[\frac{1}{RH}\, exp\left(\frac{4\sigma M_w}{RT\rho_w D_o HGF_{90}\%}\right) - 1\right] \tag{1}$$

Page 6 and Line 169: It is not clear the modified ion pairing scheme: what is the difference between SA and AS

Response: Thank you for your correction. We modified the sentence in the revised manuscript.

*(Line 178-181) "However, Gysel et al. (2007) did not include NH₄Cl in their ion-pairing scheme; therefore, we elaborated this scheme and made some modifications in this scheme to include ammonium chloride (ACl) in the calculation. Hence, our modified ion-pairing scheme includes NH₄Cl (ACl), NH₄NO₃ (AN), (NH₄)₂SO₄ (AS), NH₄HSO₄ (ABS), and H₂SO₄ (SA) are shown below:"*

Page 8 and Line 195: The author should discuss the source of the gas and meteorological data. At what height the met parameters were measured?

Response:

We sincerely thank the reviewer for pointing it out. As suggested, we have added the corresponding explanation sources of gas and meteorological data text in the revised manuscript.

*(Line 123-128) "2.2 Meteorological and Gas Data*

*The gas data was taken from the location site R.K Puram -DPCC, a continuous ambient air quality monitoring station controlled by the central control room for air quality management (Delhi-NCR). The gas data were downloaded from the CPCB website (https://app.cpcbccr.com/ccr/#/caaqm-dashboard/caaqm-landing/data). R.K. Puram is located 3.5 km northwest of IIT Delhi. The wind speed (WS), wind direction (WD), temperature (T), and relative humidity (RH) were continuously measured using an automatic weather station (Watch Dog 2000 series). The weather station is mounted over the top of the $9^{th}$-floor building of the IITD."*

Page 8 and Line 196: The author talked about PNSD. It is not clear how they measured it? Is it from the HTDMA or additionally a size spectrometer was used. A detailed schematic experimental design is needed.

Response:

Authors sincerely thank the reviewer. We think that what we discussed in the first comment's response can also be the response to this comment. However, in addition, we add statements explaining the PNSD and its measurement in the revised manuscript.

*(Line 109-110) "Particle number size distributions (PNSDs) and particle volume-size distributions (PVSDs) were measured using a Mobility Particle Size Spectrometer (MPSS (TROPOS type)).*

*(Line 158-161) 2.3.2 MPSS*

*MPSS measures electrical mobility distribution, which is then converted to PNSD in the 8 to ~800 nm mobility diameter range by applying an inversion algorithm to correct for multiple charged aerosol particles (Wiedensohler, 1988; Pfieffer et al., 2014) and diffusional losses (Wiedensohler et al., 2012; 2018)."*

Page 8 and Line 205-207: Reference is missing.

Response: We sincerely thank the reviewer for pointing it out. As suggested, we have added a reference in the text in the revised manuscript.

*(Line 233-234) "This comparatively higher ambient temperature and $O_3$ peak concentration during noontime (Fig. 3i) indicate the daytime photo-oxidation process (Nelson et al., 2023)."*

Page 8 and Line 213: It is not clear how the intensity of biomass burning activities was determined.

Response:

Authors sincerely thank the reviewer for the comment. We did not determine the intensity of the biomass-burning activities. The ambient trace gases NOx and CO are the markers of burning activities. Their concentration found a good correlation with the peak concentration of organic aerosol. Therefore, we imply that the peak in the concentration of CO and NOx indicates the local burning activities.

Page 9 and Line 220: Author should explain the nighttime peak of $SO_2$.

Response:

The authors thank the reviewer for the comment.

*(Line 244-246) "In contrast, $SO_2$ follows a different trend, with dynamic variations ranging from 0.46 to 9.55 ppb (4.41 ± 1.20) and showing peaks in the morning (09:00-12:00 hours) and at midnight (21:00-02:00 hours) associated with the local industrial stack emissions."*

Page 9 and Line 235: It is not clear about MPSS. Is it a separate instrument associated with the experimental design? If so, why was the MPSS data not presented in this study?

Response:

We sincerely thank the reviewer. We think that what we discussed in the first major comment's and $8^{th}$ minor comment's response can also be the response to this comment. However, in addition, The MPSS time series data already have been shown in the manuscript in Fig. 1(c) in terms of PNSD.

Page 9 and Line 240: … average mass concentration 46.5 ±39.6 ug/m$^3$ consistent with 112 ug/m$^3$…. This is not clear.

Response:

Thank you for your constructive comments. We modified the statement for better explanation.

*(Line 261-265) "The OA ranged between 1 and 293 (46.5 ± 39.6) µg/m$^3$ with the predominant fraction of PM$_1$, consistent with the range of 53.3 to 166 (112) µg/m$^3$ observed during winter*

*(December-February) at the present site (Gani et al., 2019). However, lower average OA concentration could be explained by the measuring period of February-March, as aerosol loading starts decreasing in February after reaching its peak in December-January (Gupta and Mandariya, 2013)."*

Page 12: The y- axis of diel Cl plot is not clear.

Response:

We sincerely thank the reviewer for pointing it out. The plot has been corrected in the revised manuscript.

*(Fig.3o: Line 303-314)*

[Figure]

*Figure 3: Diurnal variation of ambient meteorological parameters (a) % ambient relative humidity (RH), (b) temperature (T), (c) wind speed (WS), (d) wind direction (WD), and (e) particle number size distribution (PNSD), mass concentration of ambient trace gases (f) carbon mono-oxide (CO), (g) nitrogen oxides (NOx), (h) sulfur dioxide (SO₂), and (i) ozone (O₃), (j) particle volume size distribution (PVSD), mass concentration of aerosol constituents (k) organic aerosol (OA), (l) nitrate (NO₃), (m) sulfate (SO4), (n) ammonia (NH₄), and (o) chloride (Cl), mass concentration of organic aerosol species (p) more oxidized-oxygenated OA (MO-OOA), (q) less oxidized-oxygenated OA (LO-OOA), (r) biomass burning OA (BBOA), and (s) hydrocarbon like-OA (HOA), (t) geometric mean diameter of particle number size distribution (GMDPNSD) and volume fractional contribution of (u) organic aerosol (OA), (v) ammonium sulfate (AS), (w) ammonium chloride (ACl), and (x) ammonium nitrate (AN) in PM₁, and (y) geometric mean diameter of particle volume size distribution (GMDPVSD). Upper and lower boundary of shaded area represents the 95ᵗʰ and 5ᵗʰ percentile values of respective species.*

Page 17 and Line No.398: Author should explain why two linear regressions are drawn in the correlation plot (example Fig. 5a).

Response:

The light color regression lines and equations represent the correlation of all data points of $\kappa_{200nm\_90\%}$ with the volume and mass fractional contribution of ACl in $PM_1$. In contrast, the dark color regression lines and equations indicate the regression line of averaged $\kappa_{200nm\_90\%}$ over the 10% increment of ACl by volume. We add statements in the Figure caption that explain the regression lines.

*(Line 438-444)*

[Figure]

*Figure 6: Correlation plot for (a) $\kappa_{200nm\_90\%}$ vs volume fraction of ammonium chloride aerosol ($VF_{ACl}$) and (b) aerosol liquid water content (ALWC) vs mass fraction of ammonium chloride ($MF_{ACl}$). The solid circle and square marker represent the individual data points and the average of 10% volume and mass fraction increment of ACl data points, respectively. The light and dark color regression lines and equations indicate the overall and average (10% volume and mass fraction increment) correlation, respectively. The error bars indicate the standard deviation of the data points within the 10% mass and volume fractional bins.*

Page 18 and Line No.418: Author should provide the ALWC vs mass fraction of AN and AS in the supplement.

Response:

Thank you for your constructive comments. Your suggestion seems very legitimate. The plot (Fig. S8) has been incorporated in the revised manuscript.

*(Supplement, Line 76-82)*

[Figure]

[Figure]

*Figure S8: Correlation plot for (a) aerosol liquid water content (ALWC) vs mass fraction of ammonium nitrate (MFAN)* *and (b) aerosol liquid water content (ALWC) vs mass fraction of ammonium sulfate (MFAS). The solid circle and square marker represent the individual data points and the average of 10% mass fraction increment of data points, respectively. The light and dark color regression lines and equations indicate the overall and average (10% mass fraction increment) correlation, respectively. The positive error bar indicates the standard deviation of the data points within the 10% mass fractional bin.*

Page 19 and Line No 434. The author should clearly mention the dates they consider for a relatively clean period.

Response:

Thank you for your legitimate comment. We add statements explaining the clean period's date and duration in the revised manuscript.

*(Line 545-546) "The 24th and 25th of February and the 5th, 6th, and 7th of March were marked as Clean events. The night 21 hr to morning 11 hr duration was recorded as the clean duration."*

Page 22 and Line No 505-507. Is it 39% of BBOA by mass? Figure 8 is not clear. The color coding should be clarified in the plot.

Response:

Thanks. We modified the plot and add color legends in the plot to accommodate your comment.

*(Line 603-609)*

[Figure]

*Figure 9: Box plot showing variation in H-TDMA measured hygroscopic parameter of 200 nm size particles $\kappa_{H\text{-}TDMA}$ ($\kappa_{200nm\_90\%}$) in high biomass burning (H-BB), high-chloride (H-Cl), and high-hydrocarbon like organic aerosol (H-HOA) events. Different colors represent respective events in the plot. A bigger pie chart represents the overall average volume fractional contribution of various aerosol species indicated by color-coding. In addition, minor pie charts described the event average mass fractional contribution of different OA species in OA. Diffused ring color of the pie chart displays the respective event.*

Page 22 and Line No 505-507. The dates and times of the event should be clarified in the figure 1 caption.

Response:

Thanks. The dates and times of the events have been clarified from the Figure 1 caption. We add statements explaining the clean period's date and duration in the revised manuscript.

*(Line 495-497) "High BB events were noted during the initial period (1-12 February) of the field campaign. However, H-BB events were generally captured either during the midnight (01:00 hr) to morning (08:00 hr) or evening (20:00 hr) to midnight (01:00 hr). Although, sometimes, it was continued from evening (21:00 hr) to morning (11:00 hr)."*

Page 23 and Line 535: The x-axis label is missing.

Response:

We sincerely thank the reviewer for pointing it out. The plot has been corrected in the revised manuscript.

*(Line 569-571)*

[Figure]

*Figure 8: Growth Factor Probability Density Function (GF-PDF) of 20, 50, 100, 150, and 200 nm aerosol particles for the*

*(a) clean, (b) H-BB, (c) H-HOA, and H-Cl periods.*

Page 25 and Line 583: However,…time in India…This statement is not true.

Response:

Thanks. We modified the text to justify our previous statement.

*(Line 616-618) "However, we reported hygroscopicity of nucleation and Aitken mode particles using HTDMA for the first time in India."*

Supplements

Page 2 and Line 23: Author should present the time series data of MPSS during the study period.

Response:

Thanks. The MPSS time series data already have been shown in the manuscript in Fig. 1(c) in terms of PNSD.

Page 7 and Line 80: I don't see any difference in the probability distributions of BBOA, HOA and ACL. The Author should clarify it.

Response:

Thanks. Yes, the probability of potential BBOA, HOA, and ACl sources is similar. Therefore, we conclude that during H-BB events, the receptor site was influenced by air mass from some parts of Uttar Pradesh, Punjab, and Haryana comprising BBOA, HOA, and ACl aerosol.

---

## Author Comment (AC2)

**Response to the Referee (RC) 1**

Review to "Hygroscopicity of Size-Selective Aerosol Particles at Heavily Polluted Urban Atmosphere of Delhi: Impacts of Chloride Aerosol"

The authors present field measurements of size-resolved aerosol hygroscopic growth at 90% RH and bulk aerosol composition of non-refractory PM1 during wintertime in Delhi, India, and investigate the impacts of chloride on aerosol hygroscopicity and its potential to enhance aerosol-bound liquid water. The paper provides observational evidence of Ammonium Chloride as the major contributor to aerosol hygroscopic growth and liquid water content in Delhi, which highlights the role of Ammonium Chloride in aerosol-water interaction and related haze development. I would recommend publication once the following concerns are addressed.

Major comments:

1) The manuscript is a bit long and wordy to me. The authors put too much effort on the overview of the 1.5-month field measurement, and enumerate the ranges of many aerosol properties, e.g., $PM_1$ mass concentration, chemical composition mass of different species. For example, "BBOA mass concentration varied between 0.0 to 134.7 µg/m$^3$", I feel sentences like this are not as informative, and should be reduced as much as possible.

Response:

Authors thank the reviewer for highlighting this point. We have removed the sentences which were not much informative. Please see the changes in the revised manuscript (MS).

2) I strongly suggest the authors add a representative case study including major gaseous pollutants, aerosol size distribution, chemical composition, and GF-PDF of 1~2 sizes, to showcase the driving effect of $NH_4Cl$ on aerosol hygroscopicity and see if $NH_4Cl$ exists in all size ranges (i.e., 20~200, from GF-PDF).

Response:

Authors thank the reviewer for the constructive comments. Your suggestion looks very legitimate.

*However, we have specific limitations concerning aerosol chemical composition. ACSM gives only bulk chemical composition without size-resolved composition. Therefore, we chose only the most suitable 200 nm aerosol particle to look at the driving effect of $NH_4Cl$ on aerosol hygroscopicity. We have already discussed all sizes of GF-PDF and hygroscopicity in case studies like H-BB, H-Cl, H-HOA, and Clean periods. However, we think making additional case studies to incorporate your suggestion seems too repetitive here. Therefore, instead, we described major gaseous pollutants and aerosol size distribution in addition to H-BB, H-Cl,*

*H-HOA, and Clean periods in the revised manuscript in such a way that you described in your comment.*

3) As shown in Fig. 3a, the κ of 20 nm particles look quite scattered to me. For the GF-PDF of 20 nm, I am curious about how many counts of 20 nm particles were sampled for each cycle. As the counting statistics may affect the inversion of GF-PDF. According to a recent study, a total of at least 100 particle counts might be a requirement for reliable GF-PDF inversions (https://doi.org/10.5194/amt-15-2579-2022).

Response:

Authors thank the reviewer for mentioning the importance of counting statistics requirement for reliable GF-PDF inversions.

*We have gone through the above-mentioned article and checked the counting statistics and GF-PDF of 20 nm. We found that the 20 nm particle counts are less than the recommended statistics from recent article for low aerosol loading times. Authors already followed various recommended filtering processes for good scans, as mentioned in the section, and discarded significant scans. However, authors think that for region like Delhi using HTDMA instruments it will be well justified to have a slightly lesser count.*

4) As shown in Fig. 4, the diurnal variation of κ is overshadowed by the hydrophobic mode (e.g., HGF<1.2 for 100 nm). I would suggest the authors to isolate the hygroscopic mode and calculate the corresponding κ. To do so, you could either set a fixed threshold of HGF or fit the bi-modal GF-PDFs and calculate κ using the more hygroscopic mode. By doing this, the authors could probably compare the κ of the hygroscopic mode to pure $NH_4Cl$.

Response:

Thank you for your constructive comments. Your suggestion looks very legitimate.

*We calculated the κ for HGF>1.2 per the reviewer's suggestion. However, κ (0.26 ± 0.03: HGF>1.2) was not comparable to the 0.93 hygroscopicity of pure $NH_4Cl$. In addition, assuming only inorganic salts contribute to HGF>1.2, the calculated hygroscopicity was found to be 0.52 ± 0.10, higher than the observed value of 0.26 ± 0.03. It could be due to the secondary organic aerosol significantly contributing to the HGF>1.2. As we mentioned earlier, we could not calculate hygroscopicity for HGF>1.2 due to the instrument limitation.*

5) Line 351-352: The authors attribute the two-peaked pattern in the GF-PDF to daytime photochemical reactions. If that is the case, why does the HGF decrease at noontime when photochemical activities are supposed to be even stronger.

Response:

Thanks for the comment.

*We agree with the reviewer. Therefore, the primary peak (HGF < 1.2) shifted towards the higher due to the intense daytime photooxidation, as shown in Fig. 5. However, at the same time, lower inorganic contribution, especially $NH_4Cl$, and the secondary peak (HGF>1.2) shifted towards the lower HGF side, potentially responsible for the lower daytime HGF.*

6) Regarding the minor difference in $\kappa_{200nm\_90\%}$ for H-BB and H-HOA events, I doubt if that because the two events are not well separated from each other. Do the authors have a general criterion for separating the three different events, or at least show how much overlapping there is between the two events.

Response:

Thank you for your comment.

*H-BB and H-HOA were separated according to significant BBOA and HOA concentration peaks, respectively. We feel that there is no overlapping of the events as these events do not subsequently happened, as mentioned in Fig. 2. Previous studies (Chakraborty et al., 2014 and Mandariya et al., 2019) in the IGP have shown that BBOA and HOA are difficult to distinguish as they both have significant mixed signatures (mz43, 55, 57, and 60) in their mass spectra. ACSM has limitations concerning OA mass spectra as it proved only bulk m/z without high-resolution m/z fragmentation mass spectra. It could be the reason for the possible mixing of H-HOA and H-BB events. However, as mentioned in the supplementary, we followed standard protocol to separate the HOA and BBOA sources. Therefore, there is a non-significant (p>0.05) difference in hygroscopicity, possibly due to the relative changes in primary, secondary OA, and inorganic species. In the H-HOA events, the negative effect of a significantly higher fractional (41%) contribution of HOA to OA possibly balances with a positive impact of a 7% increment in secondary OA relative to H-BB.*

Chakraborty, A., Bhattu, D., Gupta, T., Tripathi, S. N. and Canagaratna, M. R.: Real-time measurements of ambient aerosols in a polluted Indian city: Sources, characteristics, and processing of organic aerosols during foggy and nonfoggy periods, J. Geophys. Res., 120(17), 9006–9019, doi:10.1002/2015JD023419, 2015.

Mandariya, A. K., Gupta, T. and Tripathi, S. N.: Effect of aqueous-phase processing on the formation and evolution of organic aerosol (OA) under different stages of fog life cycles, Atmos. Environ., 206(November 2018), 60–71, doi:10.1016/j.atmosenv.2019.02.047, 2019.

Minor comments:

1) Line 101: Use subscript in "(NH4)2SO4".

Response:

Thanks for pointing it out. As suggested, this comment has been addressed in the revised manuscript.

*(Line 99-101) "The humidity sensors positioned in the second DMA were calibrated automatically with 100 nm ammonium sulfate ((NH₄)₂SO₄) particles every 30 min at 90% RH to analyze the stability at high RH."*

2) Line 159: Wrong expression in equation (1).

Response: Thanks for pointing it out. The equation has been corrected in the revised manuscript.

*(Line 152)*

$$\kappa_{H-TDMA\_90\%} = (HGF\_90\%^3 - 1)\left[\frac{1}{RH} \, exp\left(\frac{4\sigma M_w}{RT\rho_w D_o HGF_{90}\%}\right) - 1\right] \tag{1}$$

3) Line 236: Full spell "MPSS" where it is mentioned for the first time.

Response:

Thanks. As suggested, this comment has been addressed in the revised manuscript.

*(Line 83-88 and 119-121) "Real-time atmospheric aerosol measurements were conducted simultaneously using Hygroscopic-Tandem Differential Mobility Analyzer (H-TDMA), Mobility Particle Size Spectrometer (MPSS), and Aerodyne Aerosol Chemical Speciation Monitor (ACSM, Aerodyne Research, Billerica MA) during winter (1ˢᵗ February 2020 to 16ᵗʰ March 2020) at the Indian Institute of Technology (IIT) Delhi in Block 5, at the height of nearly 15 m as shown in Fig. 1. The lab-2 is situated at the height of 15 m above the ground level and lab-1 is 50 m apart from lab-1."*

4) Line 239-241: The value of OA mass concentrations does not seem to be consistent with that reported in Gani et al., (2019).

Response:

Thank you for your constructive comments. As suggested, we modified the statement for better explanation and this comment has incorporated in the revised manuscript.

*(Line 261-265) "The OA ranged between 1 and 293 (46.5 ± 39.6) μg/m³ with the predominant fraction of PM₁, consistent with the range of 53.3 to 166 (112) μg/m³ observed during winter (December-February) at the present site (Gani et al., 2019). However, lower average OA*

*concentration could be explained by the measuring period of February-March, as aerosol loading starts decreasing in February after reaching its peak in December-January (Gupta and Mandariya, 2013)."*

5) Line 300: HGF90%_200nm of "1.12-1.179", but average to 1.41 ± 0.09?

Response:

Thanks for pointing it out. We realize that it is typo mistake. This sentence has now been corrected in the revised manuscript.

*(Line 318-322) "The hygroscopic growth factors of 20 ($HGF_{90\%\_20nm}$), 50 ($HGF_{90\%\_50nm}$), 100 ($HGF_{90\%\_100nm}$), 150 ($HGF_{90\%\_150nm}$), and 200 nm ($HGF_{90\%\_200nm}$) size particles varied between 1.00-1.41, 1.05-1.39, 1.11-1.49, 1.12-1.63, and 1.12-1.79 with an average of 1.14 ± 0.09 (average ± standard deviation), 1.16 ± 0.06, 1.27 ± 0.07, 1.35 ± 0.09, and 1.41 ± 0.09, respectively."*

6) Line 405: Full spell "ALWC" where it is mentioned for the first time.

Response:

Thanks for pointing it out. This sentence has now been modified accordingly in the revised manuscript.

*(Line 425-427) "Further, ammonium chloride has a more significant water uptake potential (Chen et al., 2022; Zhao et al., 2020), which can be justified by the solid correlation of aerosol liquid water content (ALWC) with a mass fraction of ACl in $PM_1$ as shown in Fig. 6(b)."*

Comments for figures:

1) Figure 1a and 1b: The lines are overlapped with the shaded boxes, looks like in-continuous data. Describe the shaded boxes of different colors in the caption.

Response:

Authors thank the reviewer for pointing it out. The Fig. 2 now has been corrected in the revised manuscript.

[Figure]

*Figure 2: Temporal variability of ambient (a) relative humidity (RH), temperature (T), (b) wind speed (WS), wind direction (WD), (c) particle number-size distribution (PNSD), 24-average geometric mean diameter (GMD), (d) particle volume-size distribution (PVSD), (e) particulate matter (PM₁), organic aerosol (OA), nitrate (NO₃), sulfate (SO₄), ammonium (NH₄), chloride (Cl), (f) fractional contribution of OA, NO₃, SO₄, NH₄, and Cl in PM₁, (g) more oxidized-oxygenated OA (MO-OOA), less oxidized-oxygenated OA (LO-OOA), biomass burning OA (BBOA), hydrocarbon like-OA (HOA), and (h) fractional contribution of MO-OOA, LO-OOA, BBOA, and HOA in OA. The pie chart sub-plot represents the overall average contribution of species, and the bar sub-plot represents the overall campaign average value of different species. All other species are represented with specific color coding mentioned in legends. The light green, pink, and grey color shaded vertical line indicates the high-BBOA (H-BB), high-HOA (H-HOA), and high-Cl (H-Cl) events, respectively. The discontinuity in the data points marks the missing data or non-sampling time.*

2) Figure 2o: The y-axis label is blocked.

We sincerely thank the reviewer for pointing it out. The plot has been corrected in the revised manuscript.

[Figure]

*Figure 3: Diurnal variation of ambient meteorological parameters (a) % ambient relative humidity (RH), (b) temperature (T), (c) wind speed (WS), (d) wind direction (WD), and (e) particle number size distribution (PNSD), mass concentration of ambient trace gases (f) carbon mono-oxide (CO), (g) nitrogen oxides (NOx), (h) sulfur dioxide (SO₂), and (i) ozone (O₃), (j) particle volume size distribution (PVSD), mass concentration of aerosol constituents (k) organic aerosol (OA), (l) nitrate (NO₃), (m) sulfate (SO4), (n) ammonia (NH₄), and (o) chloride (Cl), mass concentration of organic aerosol species (p) more oxidized-oxygenated OA (MO-OOA), (q) less oxidized-oxygenated OA (LO-OOA), (r) biomass burning OA (BBOA), and (s) hydrocarbon like-OA (HOA), (t) geometric mean diameter of particle number size distribution (GMDPNSD) and volume fractional contribution of (u) organic aerosol (OA), (v) ammonium sulfate (AS), (w) ammonium chloride (ACl), and (x) ammonium nitrate (AN) in PM₁, and (y) geometric mean diameter of particle volume size distribution (GMDPVSD). Upper and lower boundary of shaded area represents the 95th and 5th percentile values of respective species.*

3) Figure 2d: The diurnal WD pattern looks quite different from the wind rose plot (i.e., Figure S6). The latter suggested a negligible fraction of southerly wind.

Response:

The authors thank the reviewer for the comment. The wind directions predominantly varied from 20 to 330 degrees from the north; therefore, Fig. 2d shows the statistically average WD of nearly 180 degrees. However, in a real scenario, southerly WD has a negligible fraction.

4) Figure 3: line 332, is Fig. 3d for 150 nm?

Response:

Thanks for pointing it out. As suggested, this suggestion has been incorporated in the revised highlights.

*(Figure 4d: Line 352-353) "(d) 150 nm ($\kappa_{150nm\_90\%}$), and (e) 200 nm ($\kappa_{200nm\_90\%}$)."*

5) Figure S5: Use a consistent unit, ppb or µg m-3.

Response:

The authors thank the reviewer for the comment. Fig. S5 has been revised in the revised supplementary.

[Figure]

*Figure S5: Temporal variability in atmospheric $NO_x$, CO, and $SO_2$ gases concentrations.*

6) Figure 8: Add a legend for the pie chart here.

Response:

Thanks. We modified the plot and add color legends in the plot to accommodate your comment.

*(Line 603-609)*

[Figure]

*Figure 9: Box plot showing variation in H-TDMA measured hygroscopic parameter of 200 nm size particles $\kappa_{H\text{-}TDMA}$ ($\kappa_{200nm\_90\%}$) in high biomass burning (H-BB), high-chloride (H-Cl), and high-hydrocarbon like organic aerosol (H-HOA) events. Different colors represent respective events in the plot. A bigger pie chart represents the overall average volume fractional contribution of various aerosol species indicated by color-coding. In addition, minor pie charts described the event average mass fractional contribution of different OA species in OA. Diffused ring color of the pie chart displays the respective event.*

---

## Author Response (AR2)

Dear editor, we would like to express our gratitude to you and the two referees for providing thoughtful comments and valuable suggestions that have significantly improved the quality of our manuscript. Below, you will find our point-by-point responses to the individual comments of the two referees and editor.

**Response to the Referee (RC) 2**

Minor comments:

1. Fig. 2 caption: I thought the pink boxes indicated the high-CL events. Please double check.

Response:

Authors thank the reviewer for highlighting this point. Yes, the pink boxes indicate the H-Cl events. We have corrected the related text. Please see the changes in the revised manuscript (MS).

(Line 299-308) "Figure 2: Temporal variability of ambient (a) relative humidity (RH), temperature (T), (b) wind speed (WS), wind direction (WD), (c) particle number-size distribution (PNSD), 24-average geometric mean diameter (GMD), (d) particle volume-size distribution (PVSD), (e) particulate matter ($PM_1$), organic aerosol (OA), nitrate ($NO_3$), sulfate ($SO_4$), ammonium ($NH_4$), chloride (Cl), (f) fractional contribution of OA, $NO_3$, $SO_4$, $NH_4$, and Cl in $PM_1$, (g) more oxidized-oxygenated OA (MO-OOA), less oxidized-oxygenated OA (LO-OOA), biomass burning OA (BBOA), hydrocarbon like-OA (HOA), and (h) fractional contribution of MO-OOA, LO-OOA, BBOA, and HOA in OA. The pie chart sub-plot represents the overall average contribution of species, and the bar sub-plot represents the overall campaign average value of different species. All other species are represented with specific color coding mentioned in legends. The light green, grey, and pink color shaded vertical line indicates the high-BBOA (H-BB), high-HOA (H-HOA), and high-Cl (H-Cl) events, respectively. The discontinuity in the data points marks the missing data or non-sampling time."

2. Fig. 3d: The wind direction should be averaged arithmetically. Vector averaging should be used to do so. Check the following link for instructions: https://math.stackexchange.com/questions/44621/calculate-average-wind-direction

Response:

Authors thank the reviewer for the constructive comments. Your suggestion looks very legitimate. Authors revised the calculation as per suggestion and revised the plot. The plot has been corrected in the revised manuscript.

[Figure]

*Figure 3: Diurnal variation of ambient meteorological parameters (a) % ambient relative humidity (RH), (b) temperature (T), (c) wind speed (WS), (d) wind direction (WD), and (e) particle number size distribution (PNSD), mass concentration of ambient trace gases (f) carbon mono-oxide (CO), (g) nitrogen oxides (NOx), (h) sulfur dioxide (SO₂), and (i) ozone (O₃), (j) particle volume size distribution (PVSD), mass concentration of aerosol constituents (k) organic aerosol (OA), (l) nitrate (NO₃), (m) sulfate (SO₄), (n) ammonia (NH₄), and (o) chloride (Cl), mass concentration of organic aerosol species (p) more oxidized-oxygenated OA (MO-OOA), (q) less oxidized-oxygenated OA (LO-OOA), (r) biomass burning OA (BBOA), and (s) hydrocarbon like-OA (HOA), (t) geometric mean diameter of particle number size distribution (GMDPNSD) and volume fractional contribution of (u) organic aerosol (OA), (v) ammonium sulfate (AS), (w) ammonium chloride (ACl), and (x) ammonium nitrate (AN) in PM₁, and (y) geometric mean diameter of particle volume size distribution (GMDPVSD). Upper and lower boundary of shaded area represents the 95th and 5th percentile values of respective species.*

3. No x-axis label for Fig. 8 in the revised manuscript.

Response:

The plot has been corrected in the revised manuscript.

[Figure]

*Figure 8: Growth Factor Probability Density Function (GF-PDF) of 20, 50, 100, 150, and 200 nm aerosol particles for the (a) clean, (b) H-BB, (c) H-HOA, and H-Cl periods.*

**Response to the Referee (RC) 3**

There is a technical error still in the revised version and need to be fixed before publication. The figure 3 (o) left axis is not well adjusted.

Authors thank the reviewer for highlighting this point. Please see the changes in the revised manuscript (MS).

[Figure]

*Figure 3: Diurnal variation of ambient meteorological parameters (a) % ambient relative humidity (RH), (b) temperature (T), (c) wind speed (WS), (d) wind direction (WD), and (e) particle number size distribution (PNSD), mass concentration of ambient trace gases (f) carbon mono-oxide (CO), (g) nitrogen oxides (NOx), (h) sulfur dioxide (SO₂), and (i) ozone (O₃), (j) particle volume size distribution (PVSD), mass concentration of aerosol constituents (k) organic aerosol (OA), (l) nitrate (NO₃), (m) sulfate (SO₄), (n) ammonia (NH₄), and (o) chloride (Cl), mass concentration of organic aerosol species (p) more oxidized-oxygenated OA (MO-OOA), (q) less oxidized-oxygenated OA (LO-OOA), (r) biomass burning OA (BBOA), and (s) hydrocarbon like-OA (HOA), (t) geometric mean diameter of particle number size distribution (GMD_PNSD) and volume fractional contribution of (u) organic aerosol (OA), (v) ammonium sulfate (AS), (w) ammonium chloride (ACl), and (x) ammonium nitrate (AN) in PM₁, and (y) geometric mean diameter of particle volume size distribution (GMD_PVSD). Upper and lower boundary of shaded area represents the 95th and 5th percentile values of respective species.*

**Response to the Editor**

1. The readability and the grammar of the manuscript need to be deeply improved. It is highly recommended that the manuscript is reviewed by a native English speaker.

Authors thank the reviewer for highlighting this point. Please see the changes in the revised manuscript (MS).

2. In the abstract several terms are defined; however, they are only used once. Please only use an abbreviation when a specific term is going to be used several times. For example, there is no need to use the following abbreviations in the abstract: ACSM, DAS, H-TDMA, CCN, and OA.

Authors thank the reviewer for highlighting this point. Please see the changes in the revised manuscript (MS).

*"**Abstract.** Recent research has revealed the crucial role of winter-time, episodic high chloride (H-Cl) emissions in the Delhi region, which significantly impact aerosol hygroscopicity and aerosol-bound liquid water, thus contributing to the initiation of Delhi fog episodes. However, these findings have primarily relied on modeled aerosol hygroscopicity, necessitating validation through direct hygroscopicity measurements. This study presents the measurements of non-refractory bulk aerosol composition of $PM_1$ from an Aerodyne aerosol chemical speciation monitor and for first-time size-resolved hygroscopic growth factors (Nucleation, Aitken, and Accumulated mode particles) along with their associated hygroscopicity parameters at 90% relative humidity using a hygroscopic-tandem differential mobility analyzer at the Delhi Aerosol Supersite. Our observations demonstrate that the hygroscopicity parameter for aerosol particles varies from 0.00 to 0.11 (with an average of 0.03 ± 0.02) for 20 nm particles, 0.05 to 0.22 (0.11 ± 0.03) for 50 nm particles, 0.05 to 0.30 (0.14 ± 0.04) for 100 nm particles, 0.05 to 0.41 (0.18 ± 0.06) for 150 nm particles, and 0.05 to 0.56 (0.22 ± 0.07) for 200 nm particles. Surprisingly, our findings demonstrate that the period with H-Cl emissions displays notably greater hygroscopicity (0.35 ± 0.06) in comparison to spans marked by high biomass burning (0.18 ± 0.04), high hydrocarbon-like organic aerosol (0.17 ± 0.05), and relatively cleaner periods (0.27 ± 0.07). This research presents initial observational proof that ammonium chloride is the main factor behind aerosol hygroscopic growth and aerosol-bound liquid water content in Delhi. The finding emphasizes, ammonium chloride's role in aerosol-water interaction and related haze/fog development. Moreover, the high chloride levels in aerosols seem to prevent the adverse impact of high organic aerosol concentrations on cloud condensation nuclei activity."*

3. On the other hand, the authors define the same term several times along the manuscript. It is recommended that a term is defined the first time it is used and later on only the abbreviation is used. For example, H-Cl should be defined in L15 and not in L26.

Thank you for pointing out it. Authors modified the text throughout the manuscript. Please see the changes in the revised manuscript (MS).

4. Figures S3 and S4 are not called in the main text.

Thank you for pointing out it. Please see the changes in the revised manuscript (MS).

*(Line 206-207) "Our results shows a strong correlation and nearly unit slope (0.9999) between the calculated and modeled inorganic salts, as presented in Fig. S3."*

*(Line 264-266) "Additionally, we observed a high correlation ($r^2 = 0.83$, $p<0.05$) between $PM_1$ measured by ACSM and MPSS, assuming an effective aerosol density of 1.6 $g/cm^3$ (refer to Fig. S4)."*

5. Figures S5 and S6 cannot be called before Figures S2, S3, and S4.

Thank you for pointing out it. Please see the changes in the revised manuscript (MS).

L29-30: "Ammonium Chloride" should be "ammonium chloride"

Thank you for pointing out it. Please see the changes in the revised manuscript (MS).

*(Line 28-30) "This research presents initial observational proof that ammonium chloride is the main factor behind aerosol hygroscopic growth and aerosol-bound liquid water content in Delhi."*

L35: Add a reference after "budgets"

Thank you for pointing out it. Please see the changes in the revised manuscript (MS).

*(Line 34-36) "The Intergovernmental Panel on Climate Change (IPCC) (Intergovernmental Panel on Climate Change, 2023) reported that the interaction between aerosols and clouds is not completely comprehended, and there are significant uncertainties in gauging global radiative budgets."*

*Reference:*

*Intergovernmental Panel on Climate Change: Climate Change 2023 – The Physical Science Basis, Cambridge University Press., 2023.*

L39: "is crucial to predict the" should be "is crucial to better predict the"

Thank you for pointing out it. The txt has been modified in the revised manuscript. Please see the changes in the revised manuscript (MS)

*(Line 40-42) "Its comprehension is vital to better predicting the aerosol size distribution and scattering properties with more accuracy in global models under varying atmospheric humidity (RH) conditions (Randall et al., 2007)."*

L41: Delete "atmospheric conditions"

Thank you for pointing out it. The text has been modified in the revised manuscript. Please see the changes in the revised manuscript (MS).

*(Line 42-44)" Hygroscopicity at higher RH results in an increase in the cross-sectional area of the aerosol, leading to efficient light scattering by the aerosol particles (Tang and Munkelwitz, 1994)."*

L56: It seems that "counter" is misplaced.

Thank you for pointing out it. The text has been modified in the revised manuscript. Please see the changes in the revised manuscript (MS).

*(Line 55-59)" Over the past few decades, researchers have extensively measured aerosol hygroscopicity using a hygroscopic tandem differential mobility analyzer (H-TDMA) (Massling et al., 2005; Gysel et al., 2007; Mandariya et al., 2020; Swietlicki et al., 2008; Yeung et al., 2014; Kecorius et al., 2019) and a CCN counter (Bhattu and Tripathi, 2015; Gunthe et al., 2011; Massoli et al., 2010; Ogawa et al., 2016) under sub- and supersaturated conditions, respectively."*

L58-61. Improve the grammar.

Thank you for pointing out it. The text has been modified in the revised manuscript. Please see the changes in the revised manuscript (MS).

*(Line 55-65)" Over the past few decades, researchers have extensively measured aerosol hygroscopicity using a*

*hygroscopic tandem differential mobility analyzer (H-TDMA) (Massling et al., 2005; Gysel et al., 2007;*

*Mandariya et al., 2020; Swietlicki et al., 2008; Yeung et al., 2014; Kecorius et al., 2019) and a CCN counter*

*(Bhattu and Tripathi, 2015; Gunthe et al., 2011; Massoli et al., 2010; Ogawa et al., 2016) under sub- and*

*supersaturated conditions, respectively. Petters and Kreidenweis (2007) introduced the hygroscopicity*

*parameter, kappa (κ), to correlate aerosol hygroscopicity with its chemical composition. Hygroscopicity of OA*

*may differ according to their chemical properties such as solubility, extent of dissociation in aerosol water, and*

*surface activity, which can pose challenges in quantifying OA hygroscopicity (Hallquist et al., 2009; Jimenez et*

*al., 2009). As a result, this introduces further discrepancies in predicted and measured aerosol hygroscopicity.*

*Therefore, there is a requirement to investigate the measurement-based aerosol hygroscopicity of Delhi's*

*atmosphere to gain a better understanding of the recurring occurrences of haze and cloud formations."*

L62: "Delhi's atmosphere to understand" should be "Delhi's atmosphere to better understand"

Thank you for pointing out it. The text has been modified in the revised manuscript. Please see the changes in the revised manuscript (MS).

*(Line 62-65) "As a result, this introduces further discrepancies in predicted and measured aerosol hygroscopicity. Therefore, there is a requirement to investigate the measurement-based aerosol hygroscopicity of Delhi's atmosphere to gain a better understanding of the recurring occurrences of haze and cloud formations."*

L64-65: "to severe air quality". Do the authors mean "to severely poor/low air quality"?

Thank you for pointing out it. The text has been modified in the revised manuscript. Please see the changes in the revised manuscript (MS).

*(Line 66-67) "In recent decades, rapid economic growth and industrialization in the Indo-Gangetic Plain (IGP) have resulted in significantly poor air quality during the winter season (Wester et al., 2019)."*

L70-71: Improve the grammar.

Thank you for pointing out it. The text has been modified in the revised manuscript. Please see the changes in the revised manuscript (MS).

*(Line 72-73) "A recent study conducted in Delhi revealed that frequent high chloride events promote high levels of aerosol liquid water content under elevated humid conditions."*

L79-80: "hygroscopicity. Hence, it is essential to measure size-resolved aerosol hygroscopicity in Delhi's atmosphere and investigate its role in the context of high chloride". This is redundant, therefore, I suggest deleting it.

Thank you for pointing out it. The text has been DELETED in the revised manuscript. Please see the changes in the revised manuscript (MS).

L83-84: H-TDMA was already defined.

Thank you for pointing out it. The text has been modified in the revised manuscript. Please see the changes in the revised manuscript (MS).

*(Line 85-89) "Real-time measurements of atmospheric aerosols were conducted during winter (February 1, 2020 to March 20, 2020) at Indian Institute of Technology (IIT) Delhi, Block 5. H-TDMA, TROPOS-type Mobility Particle Size Spectrometer (MPSS), and Aerodyne Aerosol Chemical Speciation Monitor (ACSM) from Aerodyne Research in Billerica, MA were used simultaneously at a height of approximately 15 meters above ground level (a.g.l.) as depicted in Fig. 1. Lab-2 is located 50 meters away from Lab-1."*

L84: Add the model and manufacturer of the MPSS.

MPSS is built by TROPOS itself, so it does not have a model number as it is not a commercial product. It is called TROPOS type. The text has been changed in the revised manuscript. Please see the changes in the revised manuscript (MS).

*(Line 85-89) "Real-time measurements of atmospheric aerosols were conducted during winter (February 1, 2020 to March 20, 2020) at Indian Institute of Technology (IIT) Delhi, Block 5. H-TDMA, TROPOS-type Mobility Particle Size Spectrometer (MPSS), and Aerodyne Aerosol Chemical Speciation Monitor (ACSM) from Aerodyne Research in Billerica, MA were used simultaneously at a height of approximately 15 meters above ground level (a.g.l.) as depicted in Fig. 1. Lab-2 is located 50 meters away from Lab-1."*

L87: "The lab-2 is situated at the height of 15 m a.g.l. and lab-2 is 50 m apart" should be "The lab-2 is situated at the height of 15 m a.g.l. and is 50 m apart"

The text has been changed in the revised manuscript. Please see the changes in the revised manuscript (MS).

*(Line 85-89) "Real-time measurements of atmospheric aerosols were conducted during winter (February 1, 2020 to March 20, 2020) at Indian Institute of Technology (IIT) Delhi, Block 5. H-TDMA, TROPOS-type Mobility Particle Size Spectrometer (MPSS), and Aerodyne Aerosol Chemical Speciation Monitor (ACSM) from Aerodyne Research in Billerica, MA were used simultaneously at a height of approximately 15 meters above ground level (a.g.l.) as depicted in Fig. 1. Lab-2 is located 50 meters away from Lab-1."*

L99: "for aerosol and sheath respectively" should be "for aerosol flow and sheath flow, respectively."

The text has been changed in the revised manuscript. Please see the changes in the revised manuscript (MS).

*(Line 102-103) "There are two humidity sensors (Vaisala) in the system for aerosol flow and sheath flow respectively."*

L103: "Both the DMAs were". Fix it.

The text has been changed in the revised manuscript. Please see the changes in the revised manuscript (MS).

*(Line 107-108) "Size calibration for both DMAs involved the application of Latex particles of standard size, 200 nm prior to measurement."*

L110: MPSS was already defined.

The text has been changed in the revised manuscript. Please see the changes in the revised manuscript (MS).

*(Line 113-114) "Particle number size distributions (PNSDs) and particle volume-size distributions (PVSDs) were measured using a MPSS."*

L111: "A Detailed description" should be "A detailed description"

The text has been changed in the revised manuscript. Please see the changes in the revised manuscript (MS).

*(Line 165) "For a detailed account of the ACSM setup, please refer to Arub et al. (2020)."*

L111: "ACSM was" should be "The ACSM was"

The text has been changed in the revised manuscript. Please see the changes in the revised manuscript (MS).

*(Line 165-166) "The ACSM operated in a temperature-controlled laboratory at almost 0.1 lpm and 1-minute time resolution."*

L113: "ratio (m/z) m/z 10 to m/z 140" should be "ratio (m/z) 10 to m/z 140."

The text has been changed in the revised manuscript. Please see the changes in the revised manuscript (MS).

*(Line 166-167) "It was set to measure mass-to-charge ratio (m/z) from 10 to 140."*

Lines 111-118 need to be combined with Lines 165-175 in one single section.

The text has been changed in the revised manuscript. Please see the changes in the revised manuscript (MS).

*(Line 165-181) "For a detailed account of the ACSM setup, please refer to Arub et al. (2020). The ACSM operated in a temperature-controlled laboratory at almost 0.1 lpm and 1-minute time resolution. It was set to measure mass-to-charge ratio (m/z) from 10 to 140. The PM$_1$ aerosol beam was concentrated and directed towards the vaporizer at 600 °C. The flash-vaporized compounds were then ionized through impact ionization at 70 eV electrons and detected using a quadrupole mass spectrometer (Ng et al., 2011). The study employed a 200 millisecond amu$^{-1}$ scan speed and a pause setting of 125 for a sampling duration of 64 seconds to collect aerosol mass spectra using the ACSM technique. Refer to Gani et al. (2019) for comprehensive guidance on the ACSM operational procedures. For ACSM calibration and data processing, please refer to Patel et al. (2021). Positive matrix factorization (PMF) was conducted on the data, resulting in a four-factor solution: hydrocarbon-like OA (HOA), biomass burning OA (BBO), less-oxidized OA (LO-OOA), and more-oxidized OA (MO-OOA), as shown in Fig. S1. More information regarding PMF analysis can be found in section S.1 and Fig. S2 of the*

*Supplementary information. Three different events were identified based on the temporal variation of mass concentration peaks of BBOA, HOA, and Cl (see Fig. 2): 1) a high-residential or biomass burning period (H-BB); 2) a high-hydrocarbon-like OA period (H-HOA); and 3) a high-chloride period (H-Cl). Additionally, the "Clean Period" was defined as a period where the $PM_1$ loading was less than the $25^{th}$ percentiles ($\leq 38.7$ µg m$^{-3}$) for the sampling period. The event's starting and ending times were determined by the initial increase in concentration and subsequent return to the starting values as the concentration decreased."*

L167: "Patel et al., 2021." Fix it.

The text has been changed in the revised manuscript. Please see the changes in the revised manuscript (MS).

*(Line 171-172) "For ACSM calibration and data processing, please refer to Patel et al. (2021)."*

L220-223: RH, T, WD, WS, PNSD, and OA were already defined.

The text has been changed in the revised manuscript. Please see the changes in the revised manuscript (MS).

*(Line 225-227) "Fig. 2 depicts the hourly-resolved temporal changes of various meteorological parameters, including RH, T, WD, and WS, PNSD, PVSD, principal components of non-refractory $PM_1$, and OA with their corresponding fractional mass contributions."*

L226. RH and T were already defined.

The text has been changed in the revised manuscript. Please see the changes in the revised manuscript (MS).

*(Line 229-231) "The ambient RH and T vary within the range of 24.2% to 96.6% and 9.0 °C to 28.5 °C, respectively. The average values of RH and T are 56.0% ± 18.2% and 18.7 °C ± 4.2 °C, respectively."*

L243: WS was already defined.

The text has been changed in the revised manuscript. Please see the changes in the revised manuscript (MS).

From here on, I stopped checking the grammar, spelling, and punctuation. Please check out the rest of the manuscript.

The text has been changed in the revised manuscript. Please see the changes in the revised manuscript (MS).

---

## Author Response (AR3)

Dear editor, we would like to express our gratitude to you for providing thoughtful comments and valuable suggestions that have significantly improved the quality of our manuscript. Below, you will find our point-by-point responses to the individual comments of the editor.

**Response to the Editor**

Technical comments:

1. In the supplement file it is mentioned "Number of Figures: 13"; however, there are only 12 figures.

Authors thank the editor for highlighting this point. Please see the changes in the revised supplementary information.

*(Line 18) "Number of Figures: 12"*

2. Figures S12 is not called in the main text

Authors thank the editor for highlighting this point. Please see the changes in the revised manuscript (MS).

*(Line 577-579) "The clean duration was from 9 PM to 11 AM. E and S-E winds dominated the relatively clean period, but pollution was associated with calm winds, as shown in Fig. S12."*

3. Figures S5 (Line 227) and S6 (Line 238) are called before Fig. S4 (Line 267)

Thank you for pointing out it. Authors corrected the related text in the manuscript. Please see the changes in the revised manuscript (MS).

*(Line 225-227) "In addition, Fig. S4 exhibits the temporal fluctuations of atmospheric gases, specifically nitrogen oxides (NOx), carbon monoxide (CO), and sulfur dioxide (SO2). Delhi's winter climate is mainly affected by a depression caused by Western Disturbances, resulting in cold waves in the region."*

*(Line 236-238) "Fig. S5 displays the variation of WS and WD, ranging from 0.0 to 5.6 (with an average of 1.0 ± 1.0) m/s and 4.0 to 345.7 (with an average of 197.1 ± 84.4) degrees from the North, respectively."*

*(Line 243-245) "During intense biomass burning activities, ambient NOx levels reach a maximum of 421.2 ppb (58.4 ± 61.9). CO concentrations also reach maximum levels during similar periods as NOx, varying from 0.0 to 7.66 ppm (0.58 ± 0.79), as illustrated in Fig. S4."*

*(Line 263-265) "Additionally, we observed a high correlation ($r^2 = 0.83$, p<0.05) between $PM_1$ measured by ACSM and MPSS, assuming an effective aerosol density of 1.6 $g/cm^3$ (refer to Fig. S6)."*

4. Given that the order of the figures from the supplement were changed, please doble check that they are properly called in the main text.

Thank you for pointing out it. Please see the changes in the revised manuscript (MS).